# Boost-and-Skip: A Simple Guidance-Free Diffusion for Minority Generation

Soobin Um [* 1]   Beomsu Kim [* 1]   Jong Chul Ye [1]

## Abstract

Minority samples are underrepresented instances located in low-density regions of a data manifold, and are valuable in many generative AI applications, such as data augmentation, creative content generation, etc. Unfortunately, existing diffusion-based minority generators often rely on computationally expensive guidance dedicated for minority generation. To address this, here we present a simple yet powerful guidance-free approach called *Boost-and-Skip* for generating minority samples using diffusion models. The key advantage of our framework requires only two minimal changes to standard generative processes: (i) variance-boosted initialization and (ii) timestep skipping. We highlight that these seemingly-trivial modifications are supported by solid theoretical and empirical evidence, thereby effectively promoting emergence of underrepresented minority features. Our comprehensive experiments demonstrate that Boost-and-Skip greatly enhances the capability of generating minority samples, even rivaling guidance-based state-of-the-art approaches while requiring significantly fewer computations. Code is available at `https://github.com/soobin-um/BnS`.

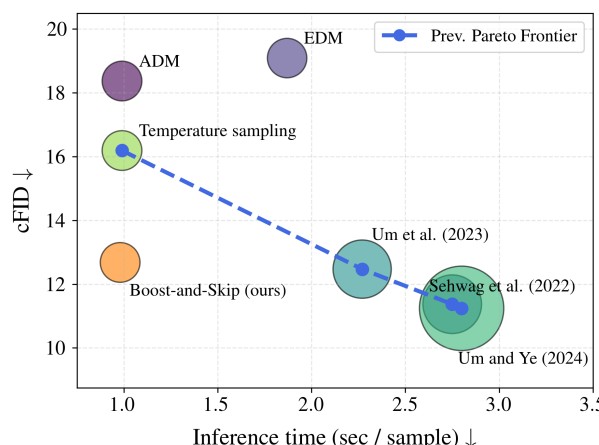

Figure 1: **Trade-off between minority generation performance and complexity on ImageNet $64 \times 64$.** The area of each bubble corresponds to peak memory consumption. cFID (Parmar et al., 2022) calculations are based on real minority data (as in Um et al. (2023); Um & Ye (2024b)), meaning that lower values indicate generation of more realistic minority samples. Note that our framework achieves competitive minority generation performance comparable to guided samplers (Sehwag et al., 2022; Um et al., 2023; Um & Ye, 2024b) while maintaining minimal complexity, thus advancing beyond the existing Pareto Frontier. See Table 3 for detailed metric values.

## 1. Introduction

Diffusion models are a prominent class of modern generative AI, known for their ability to generate high-quality content across various data modalities (Ho et al., 2020; 2022; Zhang et al., 2023). Unlike traditional frameworks like GANs (Goodfellow et al., 2014), diffusion models are notably robust, effectively learning the underlying data distribution even for rare or underrepresented examples in the training data (Sehwag et al., 2022). This inherent advantage

has been readily adopted by researchers, leading to significant progress in *minority generation* – the task of generating minority samples that reside in low-density regions of a data manifold[1] (Sehwag et al., 2022; Um et al., 2023; Um & Ye, 2024b;a). The distinctive characteristics of minority instances make them important in various applications, including medical diagnosis (Um et al., 2023), anomaly detection (Du et al., 2022; 2023), and creative AI (Rombach et al., 2022; Han et al., 2022).

Existing high-performance minority generators primarily rely upon guided sampling (Sehwag et al., 2022; Um et al., 2023; Um & Ye, 2024b;a). For instance, the authors in Se-

---

[*]Equal contribution  [1]Graduate School of AI, Korea Advanced Institute of Science and Technology (KAIST), Daejeon, Republic of Korea. Correspondence to: Jong Chul Ye <jong.ye@kaist.ac.kr>.

*Proceedings of the 42$^{nd}$ International Conference on Machine Learning*, Vancouver, Canada. PMLR 267, 2025. Copyright 2025 by the author(s).

[1]Formally, the task of minority generation is expressible as drawing instances from $\mathcal{S}_\epsilon := \{z \in \mathcal{M} : p_\theta(x) < \epsilon\}$, where $\mathcal{M}$ represents the data manifold, and $p_\theta$ denotes the implicit density learned by the diffusion model. Here $\epsilon$ is a small positive constant.

hwag et al. (2022); Um et al. (2023) employ classifier guidance (Dhariwal & Nichol, 2021) to steer generation toward low-density regions using separately trained classifiers. A more recent approach by Um & Ye (2024b) mitigates this reliance on external components (such as classifiers) by developing a self-contained sampler that derives low-density guidance solely from a pretrained diffusion model. However, this method incurs significant computational overhead due to its reliance on backpropagation through the diffusion network to compute the guidance term. Although Um & Ye (2024a) introduced a refined guidance term, specifically for text-to-image generation (Nichol et al., 2021), the computational bottleneck persists.

In this work, we depart from the paradigm of guided sampling and develop a simple and computationally efficient technique for minority generation. To this end, we first investigate potential approaches for guidance-free[2] minority sampling and highlight their pitfalls. We then present *Boost-and-Skip*, our novel guidance-free approach that sidesteps the previous issues. At a high level, our framework enables *minority-favored initialization* that encourages emergence of underrepresented features throughout generative processes without the help of additional low-density guidance. Boost-and-Skip accomplishes this through two simple modifications to standard stochastic generative processes like DDPM (Ho et al., 2020).

Specifically, we propose to initiate stochastic generation with variance-*boosted* noise instead of standard Gaussian noise to encourage initializations in low-density regions. Our second yet critical modification involves *skipping* several of the earliest timesteps to enhance the impact of low-density initialization from the first modification. We demonstrate, both theoretically and empirically, that these two modifications together lead to significantly improved minority generation, whereas each modification alone only yields marginal gains. In particular, we invoke stochastic contraction theory (Pham, 2008; Pham et al., 2009; Chung et al., 2022) to show that the contracting nature of the stochastic generative process gradually corrects unwanted spurious information introduced by the noise amplification, thereby contributing to the generation of high-quality minority samples.

To demonstrate the empirical benefits of our approach, we conducted extensive experiments across various real-world benchmarks. Importantly, our sampling technique achieves competitive minority generation performance compared to prior works (Sehwag et al., 2022; Um et al., 2023; Um & Ye,

---

[2]By *guidance-free*, we refer to approaches that do not incorporate a dedicated guidance term specifically designed to promote minority sample generation. Note that this does not preclude the use of conventional guidance mechanisms, such as classifier-free guidance (Ho & Salimans, 2022).

2024b), while offering substantially reduced computational costs. For instance, on ImageNet $64 \times 64$, our minority sampler requires 65% less wall-clock time and 4.5 times lower peak memory consumption than the current state-of-the-art method by Um & Ye (2024b). Moreover, thanks to the simplicity, our method is highly scalable. See Figure 1 for a visual illustration of these benefits and Table 3 for detailed metric values. To further highlight the practical significance of our sampler, we demonstrate its effectiveness in a potential downstream application, specifically its use in data augmentation for classification tasks.

Our contributions are summarized as follows:

- We propose *Boost-and-Skip*, a novel guidance-free approach that introduces two simple yet effective modifications to standard stochastic generative processes: (i) variance-*boosted* initialization and (ii) timestep *skipping*.
- We provide both theoretical insights and empirical evidence to validate the effectiveness of the two modifications, which together lead to significantly improved minority generation performance.
- We empirically show that our approach achieves competitive performance with substantially lower computational costs compared to state-of-the-art methods.

## 2. Background

Diffusion models are characterized by the two stochastic differential equations (SDEs): the forward and reverse SDEs (Song et al., 2020). Given a $d$-dimensional noise-perturbed data space $\boldsymbol{x}(t) \in \mathbb{R}^d$ with $t \in [0, T]$, the forward SDE is conventionally expressed as:

$$\mathrm{d}\boldsymbol{x} = \boldsymbol{f}(\boldsymbol{x}, t)\mathrm{d}t + g(t)\mathrm{d}\boldsymbol{w}, \qquad (1)$$

where $\boldsymbol{f}(\cdot, \cdot) : \mathbb{R}^d \to \mathbb{R}^d$ is a drift term, and $g(\cdot) : \mathbb{R} \to \mathbb{R}$ indicates a (scalar) diffusion coefficient coupled with a standard Wiener process $\boldsymbol{w} \in \mathbb{R}^d$. With properly chosen $\boldsymbol{f}(\cdot, \cdot)$ and $g(\cdot)$, the forward SDE describes a progressive destruction of the target data distribution $\boldsymbol{x}(0) \sim p_0$ upto a fully noised one $\boldsymbol{x}(T) \sim p_T$ that one can easily sample from (*e.g.*, a Gaussian distribution).

The reverse SDE runs backward from the noised distribution $\boldsymbol{x}(T) \sim p_T$ down to the data distribution $\boldsymbol{x}(0) \sim p_0$, formally written as (Song et al., 2020):

$$\mathrm{d}\boldsymbol{x} = \big[\boldsymbol{f}(\boldsymbol{x}, t) - g(t)^2 \nabla_{\boldsymbol{x}} \log p_t(\boldsymbol{x})\big]\mathrm{d}t + g(t)\mathrm{d}\tilde{\boldsymbol{w}}, \quad (2)$$

where $\tilde{\boldsymbol{w}}$ is a standard Wiener process with backward time flow (from $T$ to 0). The score function $\nabla_{\boldsymbol{x}} \log p_t(\boldsymbol{x})$ is commonly approximated by a neural network $\boldsymbol{s}_{\boldsymbol{\theta}}(\boldsymbol{x}, t)$ via denoising score matching (Vincent, 2011; Song et al., 2020).

One prominent instance of diffusion processes is variance-

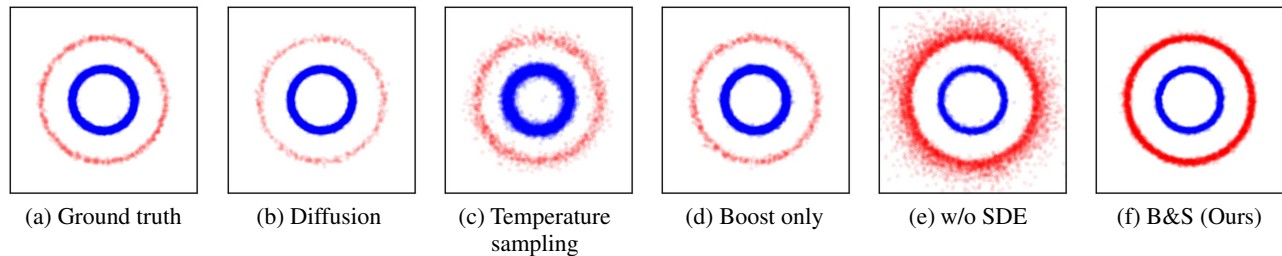

|               |               |               |               |               |               |
| :-----------: | :-----------: | :-----------: | :-----------: | :-----------: | :-----------: |
| (a) Ground truth | (b) Diffusion | (c) Temperature sampling | (d) Boost only | (e) w/o SDE | (f) B&S (Ours) |

Figure 2: A 2D distribution with two concentric circles indicated by blue and red. Red circle represents minority samples, being sampled ×10 less compared to the blue circle. **(a)** Ground truth samples. **(b)** Standard diffusion sampling assigns even less mass to minority samples. **(c)** Temperature sampling (Ackley et al., 1985) generates off-manifold samples, due to errors accumulated through sampling trajectories (see Figure 3 for details). **(d)** Boosting without skipping closely recovers the data distribution yet does not promote minority features over the ground truth. **(e)** Without stochasticity (*e.g.*, using PF-ODE), the proposed modifications fail to contract truncation error, leading to off-manifold samples. **(f)** Boost-and-Skip (B&S) generates minority samples without falling off the data manifold.

preserving (VP) SDE (Song et al., 2020):

$$\mathrm{d}\boldsymbol{x} = -\frac{1}{2}\beta(t)\boldsymbol{x}\mathrm{d}t + \sqrt{\beta(t)}\mathrm{d}\boldsymbol{w}, \qquad (3)$$

where $\beta(t)$ is a positive function which integrates to infinity to ensure $p_T \approx \mathcal{N}(\boldsymbol{0}, \boldsymbol{I})$. VP-SDE conditioned on an initial point $\boldsymbol{x}(0)$ can be characterized by a Gaussian forward diffusion kernel (Song et al., 2020):

$$p_{0t}(\boldsymbol{x}(t)|\boldsymbol{x}(0)) = \mathcal{N}\big(\boldsymbol{x}(t)|\alpha(t)\boldsymbol{x}(0), (1-\alpha(t)^2)\boldsymbol{I}\big), \quad (4)$$

where $\alpha(t)$ is defined as

$$\alpha(t) := e^{-\frac{1}{2}\int_0^t \beta(s)\mathrm{d}s}. \qquad (5)$$

The associated reverse process of VP-SDE reads:

$$\mathrm{d}\boldsymbol{x} = \left[-\frac{1}{2}\beta(t)\boldsymbol{x} - \beta(t)\nabla_{\boldsymbol{x}}\log p_t(\boldsymbol{x})\right]\mathrm{d}t + \sqrt{\beta(t)}\mathrm{d}\tilde{\boldsymbol{w}}. \qquad (6)$$

Simulating Eq. (6) with a score model $\boldsymbol{s}_{\boldsymbol{\theta}}(\boldsymbol{x}, t)$ (trained on the VP-SDE forward kernel) generates sample trajectories whose marginal distributions approximate $\{p_t(\boldsymbol{x})\}_{t=0}^T$ (Song et al., 2020); this is often called an empirical reverse VP-SDE.

**Discrete VP-SDE (DDPM).** Consider $N$ discrete points linearly distributed across timesteps $t \in [0, T]$. A discretized expression of the forward SDE in Eq. (3) is (Song et al., 2020):

$$\boldsymbol{x}_i = \sqrt{1-\beta_i}\boldsymbol{x}_{i-1} + \sqrt{\beta_i}\boldsymbol{z}_i, \quad i = 1, \dots, N, \quad (7)$$

where $\boldsymbol{z}_i \sim \mathcal{N}(\boldsymbol{0}, \boldsymbol{I})$. $\{\beta_i\}_{i=1}^N$ denotes a discretized sequence of $\beta(t)$. Unfolding Eq. (7) across discrete timesteps $i$ yields one-shot perturbation: $\boldsymbol{x}_i = \sqrt{\bar{\alpha}_i}\boldsymbol{x}_0 + \sqrt{1-\bar{\alpha}_i}\boldsymbol{z}$, where $\boldsymbol{z} \sim \mathcal{N}(\boldsymbol{0}, \boldsymbol{I})$, and $\bar{\alpha}_i := \prod_{j=1}^i \alpha_j$ for $\alpha_i := 1-\beta_i$.

By discretizing Eq. (2), the associated reverse process can be written as (Song et al., 2020):

$$\boldsymbol{x}_{i-1} = \frac{1}{\sqrt{\alpha_i}}\big\{\boldsymbol{x}_i + (1-\alpha_i)\nabla_{\boldsymbol{x}}\log p_i(\boldsymbol{x})\big\} + \sqrt{1-\alpha_i}\boldsymbol{z},$$
$$(8)$$

where $i \in \{1, \dots, N\}$ and $\nabla_{\boldsymbol{x}}\log p_i(\boldsymbol{x})$ represents the score function for discrete timesteps. Data generation is now performed by first sampling from $\boldsymbol{x}_N \sim \mathcal{N}(\boldsymbol{0}, \boldsymbol{I})$ and simulating Eq. (8) down to $\boldsymbol{x}_0$ with the learned score network $\boldsymbol{s}_{\boldsymbol{\theta}}(\boldsymbol{x}, i) \approx \nabla_{\boldsymbol{x}}\log p_i(\boldsymbol{x})$.

## 3. Method

### 3.1. Towards guidance-free minority sampler

Our framework starts by investigating potential approaches to minority-focused generation without relying on low-density guidance. One viable method is *temperature sampling* (Ackley et al., 1985), a method commonly used in traditional likelihood-based frameworks (Ackley et al., 1985; Kingma & Dhariwal, 2018). In the context of diffusion models, temperature sampling can be implemented by scaling the score function along the generation trajectory (Dhariwal & Nichol, 2021). For instance, in the empirical reverse VP-SDE that employs a pretrained score function $\boldsymbol{s}_{\boldsymbol{\theta}}(\boldsymbol{x}, t)$, this translates to:

$$\mathrm{d}\boldsymbol{x} = \left[-\frac{1}{2}\beta(t)\boldsymbol{x} - \beta(t)\frac{\boldsymbol{s}_{\boldsymbol{\theta}}(\boldsymbol{x}, t)}{\tau}\right]\mathrm{d}t + \sqrt{\beta(t)}\mathrm{d}\tilde{\boldsymbol{w}}, \quad (9)$$

where $\tau$ is the *temperature* parameter. Since $\boldsymbol{s}_{\boldsymbol{\theta}}(\boldsymbol{x}, t)/\tau \approx \nabla_{\boldsymbol{x}}\log p_t(\boldsymbol{x})^{1/\tau}$, this sampler has been regarded as producing trajectories along $\{\frac{1}{Z_t}p_t(\boldsymbol{x})^{1/\tau}\}_{t=0}^T$ (Dhariwal & Nichol, 2021), where $Z_t$ is a normalization constant. In this context, choosing $\tau > 1$ (*i.e.*, high-temperature) is expected to favor generation in low-density regions.

However, we argue that this naive application of high-temperature sampling is ineffective within the diffusion

model framework. In fact, the marginal densities of trajectories generated by Eq. (9) are distinct from $\{\frac{1}{Z_t} p_t(\boldsymbol{x})^{1/\tau}\}_{t=0}^{T}$, potentially leading to unwanted generation results. A formal description is provided below, with proof in Appendix B.1.

**Proposition 3.1.** *Consider the temperature-scaled reverse VP-SDE in Eq. (9). Assuming the score function is optimal,* i.e., $\boldsymbol{s_\theta}(\boldsymbol{x}, t) = \nabla_{\boldsymbol{x}} \log p_t(\boldsymbol{x})$, *the marginal densities of samples generated by this SDE are not equal to* $\{\frac{1}{Z_t} p_t(\boldsymbol{x})^{1/\tau}\}_{t=0}^{T}$ *in general.*

One could argue that as long as high-quality minority instances are effectively generated, it is not strictly necessary for trajectory samples to adhere to the expected marginal densities $\{\frac{1}{Z_t} p_t(\boldsymbol{x})^{1/\tau}\}_{t=0}^{T}$. However, temperature sampling presents a critical practical limitation: it often suffers from significant errors in score function estimation along the sampling trajectories, potentially leading to degraded sample quality. This is a direct consequence of the score function's scaled amplitude $(1/\tau)$, which causes intermediate samples $\boldsymbol{x}_t$ to deviate from noisy data manifold $\mathcal{M}_t$ on which the diffusion model is trained to denoise. These observations are consistent with the limited performance of temperature sampling reported in Dhariwal & Nichol (2021). See Figure 3 for our empirical analyses on this point.

Another possible solution to guidance-free minority generation is to employ the *truncation trick* (Brock et al., 2018; Karras et al., 2019) popularly employed in GANs (Goodfellow et al., 2014). This technique involves sampling the latent vector from a *truncated* normal distribution, which is often implemented via a variance-scaled Gaussian (Brock et al., 2018). However, as detailed in Section 3.3, a naive application of this technique to diffusion models yields only marginal improvements in minority sample generation; see Table 2a for empirical results.

### 3.2. Boost-and-Skip: minority-focused generation with two simple tweaks

In this section, we present *Boost-and-Skip*, a novel guidance-free approach that sidesteps the practical limitation discussed in the previous section. A key benefit of Boost-and-Skip is its simplicity combined with significant improvements and theoretical grounding. Boost-and-Skip involves two core modifications to the standard stochastic reverse diffusion process: (i) *boosting* the variance of the initial noise; and (ii) *skipping* several early timesteps.

**Variance-boosting.** Consider an empirical reverse VP-SDE implemented using the score model $\boldsymbol{s_\theta}(\boldsymbol{x}, t)$:

$$\mathrm{d}\boldsymbol{x} = \left[ -\frac{1}{2}\beta(t)\boldsymbol{x} - \beta(t)\boldsymbol{s_\theta}(\boldsymbol{x}, t) \right] \mathrm{d}t + \sqrt{\beta(t)}\mathrm{d}\tilde{\boldsymbol{w}}. \quad (10)$$

In contrast to the common ignition practice that employs

$\boldsymbol{x}(T) \sim \mathcal{N}(\boldsymbol{0}, \boldsymbol{I})$, we propose a $\gamma$-scaled initialization:

$$\hat{\boldsymbol{x}}(T) \sim \mathcal{N}(\boldsymbol{0}, \gamma^2 \boldsymbol{I}), \quad \gamma > 1, \quad (11)$$

where the range of $\gamma$ is selected to enhance minority sample generation. Intuitively, this modification encourages initializations from low-density regions of the terminal distribution $p_T$, potentially facilitating the generation of under-represented samples in the data distribution $p_0$.

However, our first modification alone, which is reminiscent of the truncation trick (used in Brock et al. (2018)), provides limited improvement. We found that this is particularly pronounced in diffusion models with a negligible terminal signal-to-noise ratio (SNR), *i.e.*, $\alpha(T) \approx 0$ for large $T$ in Eq. (5). This is because, with a near-zero terminal SNR, the influence of the low-density-encouraged initial point $\hat{\boldsymbol{x}}(T)$ diminishes due to multiplication by negligible $\alpha(T)$, resulting in a marginal effect on the generative process. We will later provide theoretical justification of this claim.

**Timestep-skipping.** We address the vanishing impact problem by skipping several of the earliest timesteps. More specifically, we start simulations of Eq. (10) from:

$$T_{\mathrm{skip}} := T - \Delta_{\mathrm{skip}}, \quad (12)$$

where $\Delta_{\mathrm{skip}}$ represents the amount of skipping, which is selected to ensure non-negligible $\alpha(T_{\mathrm{skip}})$. We found that integrating the timestep skipping with variance boosting results in a significant synergistic improvement in minority sample generation.

While employing the timestep skipping alone could often yield some performance gains, we emphasize that these improvements are limited and inferior to the combined approach; see Table 2b for details.

**Validation on toy data.** In Figure 2, we provide a sanity check of Boost-and-Skip on a two-dimensional toy dataset comprised of two concentric circles. There, we verify that variance-boosting and timestep-skipping with reverse-SDE (Figure 2(f)) is the only combination which generates on-manifold minority samples – ablating boosting, skipping, or SDE leads to inferior results. In the following section, we provide theoretical intuition as to why all three components are necessary for successful minority generation.

### 3.3. Rationale behind Boost-and-Skip

Now we delve into the underlying principles of Boost-and-Skip, elucidating why and how it is effective for minority generation. We present two key mechanisms: (i) low-density emphasis and (ii) rectification through contraction.

**Low-density emphasis.** We argue that Boost-and-Skip encourages sampling from low-density instances by amplifying their probability densities. To see this, we consider a

simple (yet non-trivial) scenario where $p_0$ follows a multivariate Gaussian distribution.

**Proposition 3.2.** *Let the data distribution be $\boldsymbol{x}(0) \sim \mathcal{N}(\boldsymbol{\mu}_0, \boldsymbol{\Sigma}_0)$, and assume the optimal score function $s_{\boldsymbol{\theta}}(\boldsymbol{x}, t) = \nabla_{\boldsymbol{x}} \log p_t(\boldsymbol{x})$ trained on $p_0$ via the forward SDE in Eq. (3). Suppose the reverse SDE in Eq. (10) is initialized with $\hat{\boldsymbol{x}}(T_{skip}) \sim \mathcal{N}(\hat{\boldsymbol{\mu}}_{T_{skip}}, \hat{\boldsymbol{\Sigma}}_{T_{skip}})$. Then, the resulting generated distribution corresponds to $\hat{\boldsymbol{x}}(0) \sim \mathcal{N}(\hat{\boldsymbol{\mu}}_0, \hat{\boldsymbol{\Sigma}}_0)$, where*

$$\hat{\boldsymbol{\mu}}_0 := \boldsymbol{\mu}_0 + \alpha(T_{skip})\boldsymbol{\Sigma}_0 \boldsymbol{\Sigma}_{T_{skip}}^{-1}(\hat{\boldsymbol{\mu}}_{T_{skip}} - \boldsymbol{\mu}_{T_{skip}}), \quad (13)$$

$$\hat{\boldsymbol{\Sigma}}_0 := \boldsymbol{\Sigma}_0 + \alpha(T_{skip})^2 \boldsymbol{\Sigma}_0^2 \boldsymbol{\Sigma}_{T_{skip}}^{-2}(\hat{\boldsymbol{\Sigma}}_{T_{skip}} - \boldsymbol{\Sigma}_{T_{skip}}). \quad (14)$$

*Here, $\boldsymbol{\mu}_{T_{skip}}$ and $\boldsymbol{\Sigma}_{T_{skip}}$ are defined as:*

$$\boldsymbol{\mu}_{T_{skip}} := \alpha(T_{skip})\boldsymbol{\mu}_0, \quad (15)$$

$$\boldsymbol{\Sigma}_{T_{skip}} := \boldsymbol{I} + \alpha(T_{skip})^2(\boldsymbol{\Sigma}_0 - \boldsymbol{I}). \quad (16)$$

See Appendix B.2 for the proof. In the considered Gaussian setting, Boost-and-Skip can be instantiated by setting $\hat{\boldsymbol{\Sigma}}_{T_{skip}} = \gamma^2 \boldsymbol{I}$ and $T_{skip} < T$. Note that when $\hat{\boldsymbol{\Sigma}}_{T_{skip}} - \boldsymbol{\Sigma}_{T_{skip}} \succ 0$, the initialization contributes to *amplify* the variance of the resulting generated distribution $\hat{\boldsymbol{\Sigma}}_0$ compared to the original $\boldsymbol{\Sigma}_0$ (see Eq. (14)). This indicates that our approach can lead to probability increases in low-density regions of $p_0$, *i.e.*, the effect of low-density emphasis.

A notable point here is that when $T_{skip} \approx T$, corresponding to the case where only boosting (*i.e.*, our first modification) is applied, the low-density emphasis impact does not manifest. This is because $\lim_{T_{skip} \to T} \alpha(T_{skip}) = \alpha(T) \approx 0$ leads to the recovery of the original data distribution $(\hat{\boldsymbol{\mu}}_0, \hat{\boldsymbol{\Sigma}}_0) \approx (\boldsymbol{\mu}_0, \boldsymbol{\Sigma}_0)$; see Eqs. (13) and (14) for details. This highlights the necessity of incorporating time-skipping for effective minority generation, and also explains why the truncation trick (used in Brock et al. (2018)) could be insufficient in the diffusion model context. The key mechanism of Boost-and-Skip can be interpreted from a signal processing viewpoint. See Appendix A.4 for detailed analyses on this perspective.

To show that the condition for the low-density emphasis effect, *i.e.*, $\hat{\boldsymbol{\Sigma}}_{T_{skip}} - \boldsymbol{\Sigma}_{T_{skip}} \succ 0$, can be satisfied in practice, we provide a corollary that characterizes the range of $T_{skip}$ over which the variance amplification impact occurs:

**Corollary 3.3.** *Suppose $\boldsymbol{\Sigma}_0 = \sigma_0^2 \boldsymbol{I}$ and $\hat{\boldsymbol{\Sigma}}_{T_{skip}} = \gamma^2 \boldsymbol{I}$, and define the quantity (if it exists)*

$$\kappa := \sqrt{(\gamma^2 - 1)/(\sigma_0^2 - 1)}.$$

*The variance-amplification effect of $\hat{\boldsymbol{\Sigma}}_0$ occurs iff*

$$T_{skip} \in \begin{cases} \varnothing & \text{if } \gamma \leq 1 \leq \sigma_0, \\ (\alpha^{-1}(\kappa), \infty) & \text{if } 1 < \gamma, \sigma_0, \\ [0, \alpha^{-1}(\kappa)) & \text{if } 1 > \gamma, \sigma_0, \\ [0, \infty) & \text{if } \sigma_0 \leq 1 \leq \gamma \text{ and } (\sigma_0, \gamma) \neq \boldsymbol{1}, \end{cases}$$

*where we define $\alpha^{-1}(\kappa) := 0$ when $\kappa > 1$.*

In Appendix A.2, we provide an illustration that exhibits the behavior of $\hat{\boldsymbol{\Sigma}}_0 := \hat{\sigma}_0^2 \boldsymbol{I}$ under the conditions specified in Corollary 3.3; see Figure 6 therein.

**Rectification via contraction.** A potential concern is whether the amplified noise components due to variance-boosted initialization may impede high-quality generation. To address this, we invoke stochastic contraction theory (Pham, 2008; Pham et al., 2009), a principle that is often used to describe error-rectifying behaviors of stochastic diffusion generative processes (Chung et al., 2022; Xu et al., 2023). Specifically under the discrete VP-SDE setting in Equation (8), we establish a general theoretical result showing that, for an arbitrary distribution $p_0$, the error introduced by the boosted initialization decays exponentially as stochastic sampling progresses. See below for a formal description of the claim.

**Proposition 3.4.** *Consider an empirical version of the discrete VP-SDE in Eq. (8):*

$$\boldsymbol{x}_{i-1} = \frac{1}{\sqrt{\alpha_i}}\{\boldsymbol{x}_i + (1 - \alpha_i)s_{\boldsymbol{\theta}}(\boldsymbol{x}_i, i)\} + \sqrt{1 - \alpha_i}\boldsymbol{z},$$

*where $i \in \{1, \ldots, N\}$. Assume the optimal score function, i.e., $s_{\boldsymbol{\theta}}(\boldsymbol{x}, i) = \nabla_{\boldsymbol{x}} \log p_i(\boldsymbol{x})$, which is trained on an arbitrary data distribution $p_0$. Consider two sample trajectories $\{\boldsymbol{x}_i\}_{i=0}^N$ and $\{\hat{\boldsymbol{x}}_i\}_{i=0}^{N_{skip}}$ where $N_{skip} < N$, which are initialized with distinct distributions: $\boldsymbol{x}_N \sim \mathcal{N}(\boldsymbol{0}, \boldsymbol{I})$ and $\hat{\boldsymbol{x}}_{N_{skip}} \sim \mathcal{N}(\boldsymbol{0}, \gamma^2 \boldsymbol{I})$. Assuming that $\{\boldsymbol{x}_i\}_{i=0}^N$ is a bounded process such that $\|\boldsymbol{x}_i\|_2 < B$ (as in Xu et al. (2023)), the expected error between samples from these two trajectories at step $i \in \{0, \ldots, N_{skip} - 1\}$ is given by:*

$$\mathbb{E}[\|\boldsymbol{x}_i - \hat{\boldsymbol{x}}_i\|_2^2] \leq \frac{2C}{1 - \lambda^2} + \lambda^{2(N_{skip}-i)}(B^2 + \gamma^2 d), \quad (17)$$

*where $\lambda$ denotes the contraction rate:*

$$\lambda := \max_{j \in \{i+1, \ldots, N_{skip}\}} \sqrt{\alpha_j}\left(\frac{1 - \bar{\alpha}_{j-1}}{1 - \bar{\alpha}_j}\right), \quad (18)$$

*and $C := d(1 - \bar{\alpha}_{N_{skip}})$.*

See Appendix B.4 for the proof. Notice that the error term $B^2 + \gamma^2 d$ decreases exponentially w.r.t. $N_{skip}$ with contraction rate $\lambda$. In Appendix A.3, we provide a more general contraction theory result that shows PF-ODE preserves (total variation) distance between distribution, whereas reverse-SDE contracts distributional distances. In other words, PF-ODE propagates initial distribution error throughout the generation process. This implies that stochasticity is critical for initial error contraction, and also explains why a naive combination of Boost-and-Skip and ODE-based samplers fail to produce high-quality minority samples.

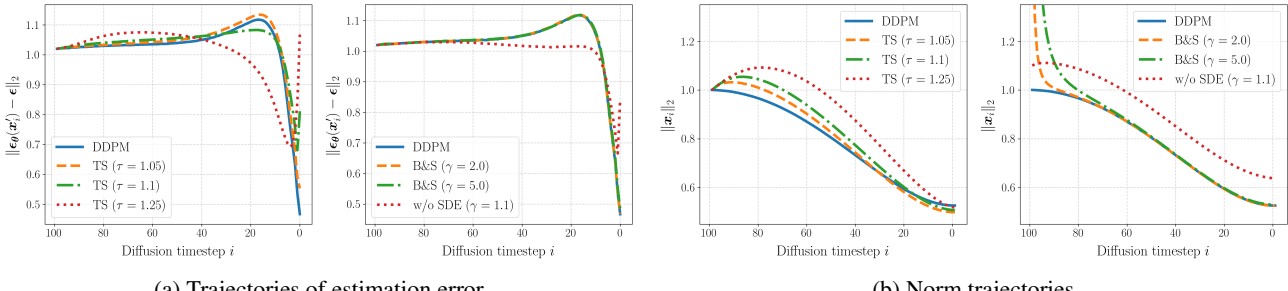

(a) Trajectories of estimation error          (b) Norm trajectories

Figure 3: **Trajectory analysis across various samplers.** "TS" refers to temperature sampling (Ackley et al., 1985), and "w/o SDE" indicates our approach without stochastic sampling, *i.e.*, using PF-ODE. "B&S" is ours, *Boost-and-Skip*. $\tau$ is the temperature parameter used to scale the score function (*e.g.*, $s_\theta(x_i, i)/\tau$), and $\gamma$ controls the boosted initialization strength. **(a)** Noise estimation errors across discrete timestep $i$, where $x_i' := \sqrt{\bar{\alpha}_i}\hat{x}_{0|i} + \sqrt{1 - \bar{\alpha}_i}\epsilon$. Here $\hat{x}_{0|i}$ represents a denoised sample from $x_i$, *i.e.*, the posterior mean of $x_i$. While temperature sampling fails even with slight variations in $\tau$, Boost-and-Skip performs as well as DDPM, demonstrating its robustness to imperfect score models. **(b)** Norm of intermediate samples $x_i$ across timestep $i$. We see that norm trajectories of our approach rapidly converge to that of DDPM, exhibiting the contraction effect claimed in Section 3.3. In contrast, trajectories of temperature sampling diverge from the DDPM curve, further revealing its pathology. See Figure 4 for visualizations of intermediate samples over trajectories.

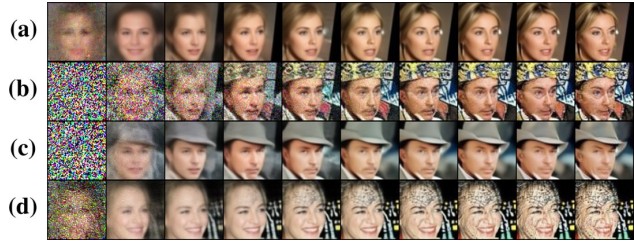

Figure 4: **Denoised intermediate samples along generative trajectories on CelebA (Liu et al., 2015). (a)** DDPM. **(b)** Temperature sampling with $\tau = 1.1$. **(c)** Ours with $\gamma = 5.0$. **(d)** Ours with $\gamma = 1.1$ and PF-ODE. From left to right, discrete timestep $i$ decreases from $i = 100$ to $0$. Leveraging the contracting nature of stochastic generation, Boost-and-Skip reduces excessive noise introduced by boosted initialization during early stages of generation.

In practice, the score function $s_\theta(x, t)$ is not optimal and may introduce estimation error. This is particularly relevant during early stages of our generative process, where the score model is tasked with estimating boosted-noise samples that were not encountered during training. Nonetheless, our empirical findings indicate that within reasonable ranges of $\gamma$, the contraction effect inherent in stochastic sampling rapidly corrects initial error during the early stages of generation. We highlight that this is in stark contrast to temperature sampling, which is highly sensitive to the choice of $\tau$ (*i.e.*, the scaling parameter) and prone to collapse even with small deviations from $\tau = 1$. Figures 3 and 4 provide empirical analyses that investigate the robustness of our approach to imperfect score models; see details therein.

## 4. Experiments

**Datasets and pretrained models.** Our experiments were conducted on four benchmarks settings with varying resolutions: (i) CelebA $64 \times 64$ (Liu et al., 2015); (ii) LSUN-Bedrooms $256 \times 256$ (Yu et al., 2015); (iii) ImageNet $64 \times 64$ (Deng et al., 2009); and (iv) ImageNet $256 \times 256$. We consider unconditional generation on CelebA and LSUN-Bedrooms, while performing class-conditional generation on the ImageNet settings. The CelebA pretrained model was developed in-house, adhering to the configuration described in Um et al. (2023). The pretrained models for LSUN-Bedrooms and ImageNet were taken from the checkpoints provided by Dhariwal & Nichol (2021).

**Baselines.** We evaluate our approach against a diverse array of baseline methods, including frameworks beyond diffusion models and minority samplers. Specifically, we include two general-purpose GAN frameworks for comparison: BigGAN (Brock et al., 2018) and StyleGAN (Karras et al., 2019). Additionally, we incorporate prominent diffusion-based approaches with their standard sampling techniques: ADM (Dhariwal & Nichol, 2021), LDM (Rombach et al., 2022), EDM (Karras et al., 2022), and DiT (Peebles & Xie, 2022). The state-of-the-art diversity sampler, CADS (Sadat et al., 2023), is also included for benchmarking. For minority sampling baselines, we consider the most advanced diffusion-based approaches: (i) Sehwag et al. (2022); (ii) Um et al. (2023); (iii) ADM-ML (Um et al., 2023) (iv) Um & Ye (2024b); and (v) temperature sampling.

**Evaluation Metrics.** We employ a set of quality and diversity metrics to evaluate generated samples. Specifically, we use: (i) Clean Fréchet Inception Distance (cFID) (Parmar et al., 2022); (ii) Spatial FID (sFID) (Nash et al., 2021);

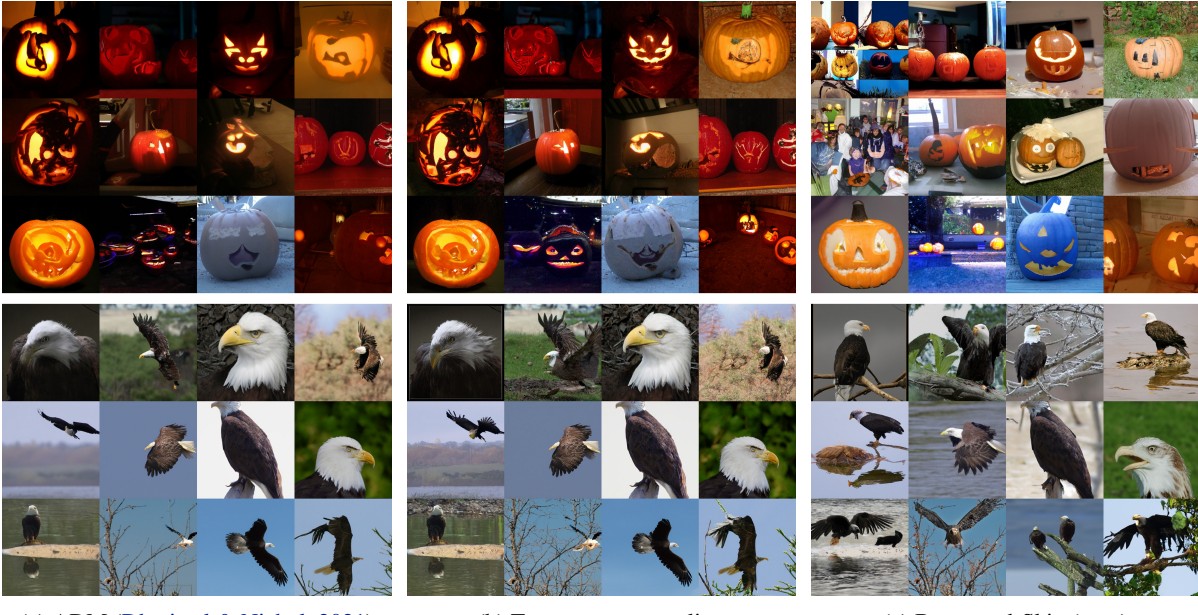

(a) ADM (Dhariwal & Nichol, 2021)     (b) Temperature sampling     (c) Boost-and-Skip (ours)

Figure 5: Sample comparison on ImageNet $256 \times 256$. Generated samples from two classes are exhibited: "jack-o'-lantern" (top row) and "bald eagle" (bottom row). We share the same random seed across all three approaches.

and (iii) Improved Precision & Recall (Kynkäänniemi et al., 2019). To evaluate the proximity to real minority data, we follow the approach used in previous works Um et al. (2023); Um & Ye (2024b). Specifically, we employ instances with the lowest likelihoods (i.e., those with the highest AvgkNN values) as reference real data for calculation of quality and diversity metrics. Additionally, to compare the capability to generate low-density instances, we adopt three uniqueness measures: (i) Average k-Nearest Neighbor (AvgkNN); (ii) Local Outlier Factor (LOF) (Breunig et al., 2000); and (iii) Rarity Score (Han et al., 2022). For all these measures, higher value indicate that an instance is less similar to its neighborhood (Sehwag et al., 2022; Han et al., 2022).

### 4.1. Results

**Qualitative comparisons.** Figure 5 presents generated samples from three different approaches on ImageNet $256 \times 256$. We observe that our framework consistently produces samples with highly more distinct and intricate visual aspects compared to the baselines, characteristics often associated with low-density instances (Serrà et al., 2019; Arvinte et al., 2023). We highlight that this contrasts with temperature sampling yielding marginal changes from the baseline ADM, as further confirmed by our quantitative evaluations (see Table 1). Additional samples including those from other benchmarks and baselines are provided in Appendix D.2.

**Quantitative evaluation.** Table 1 compares performance in terms of quality and diversity. Observe that our sampler achieves high-quality minority generating performance on

par with computationally intensive guided samplers across all datasets. We emphasize that these substantial improvements are achievable with minimal computational overhead; see Table 3 for a complexity analysis on ImageNet $64 \times 64$. In addition to its competitive performance in quality and diversity, our approach also performs well in neighborhood density metrics; see Appendix D.1 for explicit details.

**Ablation studies.** Table 2 presents ablation results on our design choices, $\gamma$ and $\Delta_t$. We see that applying either modification individually results in limited improvements (see Tables 2a and 2b). In contrast, combining both techniques with properly chosen $\gamma$ values yields significant enhancements in minority generation; see Table 2c for details. In Appendix A.5, we present a further ablation exploring performance with $\gamma < 1.0$, where we demonstrate a quality-improving potential of ours; see Table 6 for details.

**Downstream application.** To further highlight the practical importance of Boost-and-Skip, we investigate its potential application in classifier training with synthetically augmented datasets. Specifically, we examine whether our minority-promoted generated samples can enhance classification performance. Following Um & Ye (2024b), we consider the prediction of 40 attributes in CelebA and train ResNet-18 models on four datasets: (i) the CelebA training set; (ii) CelebA augmented with 50K samples from ADM (Dhariwal & Nichol, 2021); (iii) CelebA augmented with 50K samples from Um & Ye (2024b); and (iv) CelebA augmented with 50K samples from ours. As in Um & Ye (2024b), we use an off-the-shelf classifier to label the gener-

| Method | cFID ↓ | sFID ↓ | Prec ↑ | Rec ↑ | Method | cFID ↓ | sFID ↓ | Prec ↑ | Rec ↑ |
|---|---|---|---|---|---|---|---|---|---|
| **CelebA 64×64** | | | | | **LSUN Bedrooms 256×256** | | | | |
| ADM (Dhariwal & Nichol, 2021) | 75.41 | 17.11 | **0.97** | 0.23 | ADM (Dhariwal & Nichol, 2021) | 63.30 | 8.00 | 0.89 | 0.15 |
| BigGAN (Brock et al., 2018) | 80.58 | 16.80 | **0.97** | 0.19 | LDM (Rombach et al., 2022) | 63.53 | 7.73 | **0.90** | 0.13 |
| | | | | | StyleGAN (Karras et al., 2019) | 57.17 | 7.78 | 0.89 | 0.14 |
| ADM-ML (Um et al., 2023) | 51.99 | 13.40 | 0.94 | 0.30 | Um et al. (2023) | 41.75 | 7.26 | 0.87 | 0.10 |
| Sehwag et al. (2022) | 28.25 | 10.64 | 0.82 | 0.42 | Um & Ye (2024b) | **36.94** | **5.13** | 0.87 | 0.15 |
| Um et al. (2023) | 27.32 | 8.66 | 0.89 | 0.33 | | | | | |
| Um & Ye (2024b) | **19.34** | 8.85 | 0.82 | 0.47 | Temperature sampling | 51.81 | 8.22 | 0.82 | **0.21** |
| | | | | | Boost-and-Skip (proposed) | 44.77 | 6.70 | 0.78 | 0.20 |
| Temperature sampling | 37.78 | 14.49 | 0.79 | 0.38 | | | | | |
| Boost-and-Skip (proposed) | 19.79 | **6.87** | 0.77 | **0.51** | **ImageNet 256×256** | | | | |
| | | | | | ADM (Dhariwal & Nichol, 2021) | 13.22 | 7.66 | **0.86** | 0.39 |
| **ImageNet 64×64** | | | | | DiT (Peebles & Xie, 2022) | 21.51 | 6.76 | 0.80 | 0.46 |
| ADM (Dhariwal & Nichol, 2021) | 18.37 | 5.39 | 0.79 | 0.53 | CADS (Sadat et al., 2023) | 15.95 | 6.18 | 0.81 | **0.48** |
| EDM (Karras et al., 2022) | 19.09 | 4.73 | 0.73 | 0.59 | Sehwag et al. (2022) | 10.93 | 6.66 | 0.85 | 0.39 |
| Sehwag et al. (2022) | 11.37 | 4.69 | **0.80** | 0.52 | Um et al. (2023) | 11.44 | 4.63 | 0.85 | 0.42 |
| Um et al. (2023) | 12.47 | **3.13** | 0.76 | 0.56 | Um & Ye (2024b) | **9.98** | 4.35 | 0.83 | 0.45 |
| Um & Ye (2024b) | **11.24** | 3.17 | 0.73 | **0.62** | | | | | |
| | | | | | Temperature sampling | 12.48 | 6.72 | 0.84 | 0.41 |
| Temperature sampling | 16.19 | 4.20 | 0.76 | 0.58 | Boost-and-Skip (proposed) | 10.04 | **4.33** | 0.83 | 0.45 |
| Boost-and-Skip (proposed) | 12.68 | 3.18 | 0.74 | 0.61 | | | | | |

Table 1: **Quantitative comparisons.** "ADM-ML" represents a classifier-guided baseline that implements conditional generation on **M**inority **L**abels (Um et al., 2023). For baseline real data to compute the metrics, we employ the most unique samples that yield the highest AvgkNN values, following the previous convention (Um et al., 2023; Um & Ye, 2024b). The best results are marked in **bold**, and the second bests are underlined.

| $\gamma^2$ | cFID ↓ | sFID ↓ | Prec ↑ | Rec ↑ |
|---|---|---|---|---|
| 1.0 | 84.98 | 23.07 | **0.98** | 0.14 |
| 4.0 | **81.75** | **22.71** | 0.97 | 0.17 |
| 9.0 | 82.10 | 22.78 | **0.98** | 0.15 |
| 16.0 | 83.32 | 22.91 | **0.98** | 0.18 |

(a) Boost-only (*i.e.*, $\Delta_t = 0$)

| $\Delta_t$ | cFID ↓ | sFID ↓ | Prec ↑ | Rec ↑ |
|---|---|---|---|---|
| 0 | 84.98 | 23.07 | **0.98** | **0.14** |
| 10 | 61.86 | **19.97** | 0.96 | 0.13 |
| 20 | **55.15** | 22.02 | 0.91 | 0.06 |
| 50 | 289.70 | 119.59 | 0.29 | 0.00 |

(b) Skip-only (*i.e.*, $\gamma = 1.0$)

| $\gamma^2$ | cFID ↓ | sFID ↓ | Prec ↑ | Rec ↑ |
|---|---|---|---|---|
| 1.0 | 74.41 | 20.50 | **0.97** | 0.21 |
| 2.0 | 51.47 | 16.43 | 0.94 | 0.33 |
| 4.0 | **23.56** | **12.17** | 0.77 | 0.50 |
| 9.0 | 219.21 | 45.59 | 0.05 | **0.67** |

(c) Boost-and-Skip (with $\Delta_t = 3$)

Table 2: **Exploring the design space of Boost-and-Skip.** "Boost-only" refers to using boosted initialization alone, while "Skip-only" represents configurations that only employ the proposed timestep skipping. $\gamma$ denotes the boosting scale, and $\Delta_t$ indicates the number of timesteps skipped. Using either technique individually results in limited performance improvements.

| | | | Complexity | | |
|---|---|---|---|---|---|
| Method | cFID ↓ | sFID ↓ | Infer ↓ | Extra ↓ | Memory ↓ |
| ADM (Dhariwal & Nichol, 2021) | 18.37 | 5.39 | 0.99 s | – | 85.73 MB |
| EDM (Karras et al., 2022) | 19.09 | 4.73 | 1.87 s | – | 84.48 MB |
| Sehwag et al. (2022) | 11.37 | 4.69 | 2.75 s | > 16 d | 187.04 MB |
| Um et al. (2023) | 12.47 | **3.13** | 2.27 s | > 16 d | 184.40 MB |
| Um & Ye (2024b) | 11.24 | 3.17 | 2.80 s | – | 386.75 MB |
| Temperature sampling | 16.19 | 4.20 | 0.99 s | – | 85.73 MB |
| Boost-and-Skip (ours) | 12.68 | 3.18 | **0.98 s** | – | 85.73 MB |

Table 3: **Comparison of complexity across existing samplers on ImageNet 64 × 64.** "Infer" represents the inference time measured in seconds/sample. "Extra" refers to the additional time, in **days**, required to construct external classifiers used for guided sampling (Sehwag et al., 2022; Um et al., 2023). "Memory" indicates the peak memory usage, measured in MB/sample. All measurements are based on a single NVIDIA A100 GPU.

ated samples. Our results show classification improvements comparable to Um & Ye (2024b), despite our method being significantly more computationally efficient, further demonstrating its utility in downstream tasks.

## 5. Conclusion

We introduced *Boost-and-Skip*, a simple yet impactful guidance-free approach for minority sample generation. By

| Training data | Acc ↑ | F1 ↑ | Prec ↑ | Rec ↑ |
|---|---|---|---|---|
| CelebA trainset | 0.898 | 0.746 | 0.815 | 0.710 |
| + ADM gens (50K) | 0.897 | 0.742 | 0.808 | 0.711 |
| + SGMS gens (50K) | **0.903** | **0.757** | **0.822** | **0.724** |
| + Ours gens (50K) | 0.902 | 0.755 | 0.819 | 0.723 |

Table 4: **Data augmentation for downstream classification tasks.** "SGMS" refers to the approach by Um & Ye (2024b). All settings were evaluated on the CelebA testset and averaged over three different runs.

incorporating variance-*boosted* initialization and timestep *skipping*, our framework promotes the emergence of underrepresented features without relying on additional low-density guidance. Theoretical and empirical results validated its effectiveness, demonstrating competitive performance while significantly reducing computational costs. We further demonstrated its practical utility in downstream tasks such as data augmentation, highlighting its broader applicability in generative modeling.

A limitation is that as discussed in Section 3.3, a naive application to deterministic samplers may fail due to the absence of a contracting effect. A promising future direction is to address this limitation, extending the effectiveness of our approach to a wider range of generative processes.

## Impact Statement

One potential negative impact is the intentional misuse of our sampler to suppress the generation of minority-featured samples. This could be achieved by setting $\gamma < 1.0$ with $\Delta_t > 0$, potentially biasing the initialization toward high-density regions (as explored in Appendix A.5). Recognizing and mitigating this risk is essential, emphasizing the importance of responsible deployment to ensure inclusivity in generative modeling.

## Acknowledgments

This work was supported by the National Research Foundation of Korea under Grant RS-2024-00336454 and the Institute of Information & Communications Technology Planning & Evaluation (IITP) grant funded by the Korea government (MSIT) (RS-2025-02304967, AI Star Fellowship (KAIST)).

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

# A. Additional Discussions, Ablations, and Analyses

## A.1. Related work

In addition to closely related studies mentioned in Section 1 (Sehwag et al., 2022; Um et al., 2023; Um & Ye, 2024b;a), several other works explore distinct conditions and scenarios in the context of minority generation (Yu et al., 2020; Lin et al., 2022; Qin et al., 2023; Huang & Jafari, 2023; Samuel et al., 2023). One instance is Qin et al. (2023) wherein the authors develop a training technique to mitigate a class imbalance issue when constructing class-conditional diffusion models. A distinction w.r.t. ours is that they require a specialized training, and their method is limited to conditional diffusion models. Another notable study is done by Samuel et al. (2023). Specifically using text-to-image (T2I) diffusion models (Rombach et al., 2022), they develop a sampler to faithfully generate samples of unique text-prompts by employing real reference data instances associated with the given prompts. The key distinction to ours is that their method is limited to T2I models and rely upon a set of real reference data.

A related yet distinct line of research is to improve the diversity of conventional diffusion samplers (Sadat et al., 2023; Corso et al., 2023; Lu et al., 2024). For instance, Sadat et al. (2023) propose a simple conditioning technique to boost-up the diversity by introducing time-scheduled noise perturbations in the conditional embedding space. The difference from our study is that their method is confined to conditional diffusion models and not specifically designed for minority generation. The approaches in Corso et al. (2023); Lu et al. (2024) share similar spirits, repelling intermediate latent samples within an inference batch to produce visually distinct outputs. However, as with Um & Ye (2024b;a), these methods often introduce computational challenges, *e.g.*, due to the reliance on backpropagation.

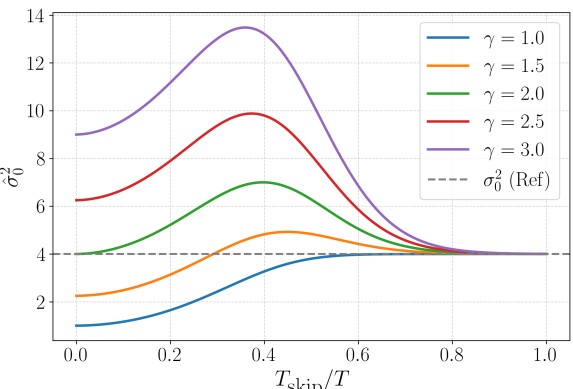

Figure 6: **Low-density emphasis impact of Boost-and-Skip.** We visualize $\hat{\sigma}_0^2$ (*i.e.*, the scale of $\hat{\Sigma}_0 \coloneqq \hat{\sigma}_0^2 I$) across $T_{\text{skip}}/T$ under the settings specified in Corollary 3.3, with $\sigma_0 = 2$. Observe that the variance of $\hat{\Sigma}_0$ surpasses that of $\Sigma_0$ for $\gamma > 1$ and $T_{\text{skip}} < T$, demonstrating the low-density encouraging influence of the Boost-and-Skip approach.

## A.2. Illustration of the impact of low-density emphasis

We continue from Section 3.3 and provide a visualization on the effect of low-density emphasis of Boost-and-Skip. Let us first recall Corollary 3.3:

**Corollary 3.3.** *Suppose $\Sigma_0 = \sigma_0^2 I$ and $\hat{\Sigma}_{T_{skip}} = \gamma^2 I$, and define the quantity (if it exists)*

$$\kappa \coloneqq \sqrt{(\gamma^2 - 1)/(\sigma_0^2 - 1)}.$$

*The variance-amplification effect of $\hat{\Sigma}_0$ occurs iff*

$$T_{skip} \in \begin{cases} \varnothing & \text{if } \gamma \leq 1 \leq \sigma_0, \\ (\alpha^{-1}(\kappa), \infty) & \text{if } 1 < \gamma, \sigma_0, \\ [0, \alpha^{-1}(\kappa)) & \text{if } 1 > \gamma, \sigma_0, \\ [0, \infty) & \text{if } \sigma_0 \leq 1 \leq \gamma \text{ and } (\sigma_0, \gamma) \neq \mathbf{1}, \end{cases}$$

*where we define $\alpha^{-1}(\kappa) \coloneqq 0$ when $\kappa > 1$.*

Figure 6 illustrates the behavior of $\hat{\Sigma}_0 \coloneqq \hat{\sigma}_0^2 I$ under the conditions specified in Corollary 3.3, where $\sigma_0 = 2$. We see that the variance scale of $\hat{\Sigma}_0$ exceeds that of $\Sigma_0$ for $\gamma > 1$ and $T_{\text{skip}} < T$, demonstrating the variance-boosting effect induced by the proposed modifications. Notably, regardless of the value of $\gamma$, this amplification effect does not occur for $T_{\text{skip}} \approx T$, thus supporting the effectiveness of our time-skipping technique for promoting low-density emphasis.

## A.3. Further results on contraction theory

We continue from Section 3.3 and extend our analysis of contraction theory in the context of Boost-and-Skip. To this end, we first explore its behavior in non-stochastic generative processes that lack the contraction effect. Specifically, we consider

its application to the probability flow ODE (PF-ODE) (Song et al., 2020) associated with Eq. (10):

$$d\boldsymbol{x} = \left[ -\frac{1}{2}\beta(t)\boldsymbol{x} - \frac{1}{2}\beta(t)\boldsymbol{s}_{\boldsymbol{\theta}}(\boldsymbol{x}, t) \right] dt. \tag{19}$$

The following proposition characterizes the density of generated samples $\hat{p}_{\text{ODE}}$ when going through the above ODE under the same settings as Proposition 3.2:

**Proposition A.1.** *Consider the same data distribution $p_0$ and the optimal score function $\boldsymbol{s}_{\boldsymbol{\theta}}(\boldsymbol{x}, t)$ as Proposition 3.2. Suppose the PF-ODE in Eq. (19) is initialized with $\hat{\boldsymbol{x}}(T_{skip}) \sim \mathcal{N}(\hat{\boldsymbol{\mu}}_{T_{skip}}, \hat{\boldsymbol{\Sigma}}_{T_{skip}})$. Then, the resulting distribution at $t = 0$ is given by $\hat{\boldsymbol{x}}_{ODE}(0) \sim \mathcal{N}(\hat{\boldsymbol{\mu}}_{0,ODE}, \hat{\boldsymbol{\Sigma}}_{0,ODE})$, where*

$$\hat{\boldsymbol{\mu}}_{0,ODE} := \boldsymbol{\mu}_0 + \boldsymbol{\Sigma}_0^{1/2} \boldsymbol{\Sigma}_{T_{skip}}^{-1/2} (\hat{\boldsymbol{\mu}}_{T_{skip}} - \boldsymbol{\mu}_{T_{skip}}), \tag{20}$$

$$\hat{\boldsymbol{\Sigma}}_{0,ODE} := \boldsymbol{\Sigma}_0 \boldsymbol{\Sigma}_{T_{skip}}^{-1} \hat{\boldsymbol{\Sigma}}_{T_{skip}}. \tag{21}$$

*Here, $\boldsymbol{\mu}_{T_{skip}}$ and $\boldsymbol{\Sigma}_{T_{skip}}$ are defined as in Eqs. (15) and (16), respectively.*

We leave the proof in Appendix B.5. We see in Eq. (21) that the variance $\hat{\boldsymbol{\Sigma}}_0$ is multiplied by the initial variance $\boldsymbol{\Sigma}_{T_{skip}}$. This implies that the low-density emphasis impact observed in the SDE-case may also manifest in the considered ODE case. However, the mean and variance formulas of the ODE-generated distribution $\hat{p}_{\text{ODE}}$ lack the $\alpha(T_{\text{skip}})$-multiplication that contributes to the recovery of $p_0$, indicating the inability of (non-stochastic) ODEs for error rectification. Below, we present a weaker but more general stochastic contraction result which shows that a similar phenomenon occurs with VP-SDE with general data distributions.

**Proposition A.2** (Error contraction for VP-SDE diffusion models)**.** *Assume the same setup as Theorem 3 in Xu et al. (2023), i.e., $\|t\nabla \log p_t(\boldsymbol{x})\| \leq L_1$ and $\|\boldsymbol{x}_t\| < B/2$ for any $\boldsymbol{x}_t$ in the support of $p_t$ and reverse-SDE trajectories. Let*

$$\boldsymbol{x}_{t_{\min}}^{\text{SDE}} = \text{SDE}(\boldsymbol{x}_{t_{\max}}, t_{\max} \to t_{\min}), \ \boldsymbol{y}_{t_{\min}}^{\text{SDE}} = \text{SDE}(\boldsymbol{y}_{t_{\max}}, t_{\max} \to t_{\min}) \tag{22}$$

$$\boldsymbol{x}_{t_{\min}}^{\text{ODE}} = \text{ODE}(\boldsymbol{x}_{t_{\max}}, t_{\max} \to t_{\min}), \ \boldsymbol{y}_{t_{\min}}^{\text{ODE}} = \text{ODE}(\boldsymbol{y}_{t_{\max}}, t_{\max} \to t_{\min}) \tag{23}$$

*where SDE and ODE denote solutions to the reverse-SDE and PF-ODE, respectively. Then*

$$\text{TV}(\boldsymbol{x}_{t_{\min}}^{\text{ODE}}, \boldsymbol{y}_{t_{\min}}^{\text{ODE}}) = \text{TV}(\boldsymbol{x}_{t_{\max}}, \boldsymbol{y}_{t_{\max}}) \tag{24}$$

$$\text{TV}(\boldsymbol{x}_{t_{\min}}^{\text{SDE}}, \boldsymbol{y}_{t_{\min}}^{\text{SDE}}) \leq \left( 1 - 2Q\left( \frac{B}{2\sqrt{\alpha(t_{\max})^{-2} - \alpha(t_{\min})^{-2}}} \right) \cdot e^{-BL_1/t_{\min} - L_1^2 \alpha(t_{\max})^{-2}/t_{\min}^2} \right) \text{TV}(\boldsymbol{x}_{t_{\max}}, \boldsymbol{y}_{t_{\max}}) \tag{25}$$

*where $Q(r) := \mathbb{P}(\varepsilon \geq r)$ for $\varepsilon \sim \mathcal{N}(0, 1)$.*

The proof can be found in Appendix B.6. Here, TV denotes the total variation distance. Indeed, we observe that PF-ODE does not contract any initial error. On the other hand, the multiplicative factor on the RHS of Eq. (25) shows that reverse-SDE reduces initial error. We empirically found that this absence of the rectifying capability often makes ODE-based samplers struggle to generate high-quality minority samples with Boost-and-Skip. See Figures 2, 3 and 4 for instance.

## A.4. How Boost-and-Skip works: a signal processing perspective

We continue from Section 3.3 and investigate the principle of Boost-and-Skip in a signal processing viewpoint. As explored in Lin et al. (2024); Hu et al. (2024), generated samples from diffusion models are often affected by initial Gaussian noise, particularly in shaping low-frequency components of images (like brightness). This effect is especially pronounced in diffusion frameworks that employ noise schedules with non-zero terminal SNR (*i.e.*, $\alpha(T) \not\approx 0$), allowing low-frequency components of the initial noise to propagate into the generative process.

From this standpoint, Boost-and-Skip can be viewed as a two-step approach. First, the low-frequency components of the noise are amplified through variance *boosting* (by Plancherel theorem (Plancherel & Leffler, 1910)), enhancing the visual variability of the random Gaussian noise. Second, this amplified low-frequency information is made more influential by initiating the generative process at a *skipped* initial timestep $T_{\text{skip}}$ with a non-zero terminal SNR ($\alpha(T_{\text{skip}}) \not\approx 0$). We conducted an empirical analysis to support this perspective. See Table 5 for explicit details.

## A.5. Extended ablation: effect of using $\gamma < 1.0$

| $f_{\text{cutoff, LPF}}$ | cFID ↓ | sFID ↓ | Prec ↑ | Rec ↑ |
|---|---|---|---|---|
| 64 (uncut) | 23.56 | 12.17 | 0.77 | 0.50 |
| 48 | 23.38 | 12.31 | 0.77 | 0.50 |
| 32 | 23.82 | 12.32 | 0.77 | 0.51 |
| 16 | 28.46 | 14.34 | 0.84 | 0.49 |
| 8 | 33.08 | 14.98 | 0.84 | 0.44 |

(a) Low-pass filtered after boosting

| $f_{\text{cutoff, HPF}}$ | cFID ↓ | sFID ↓ | Prec ↑ | Rec ↑ |
|---|---|---|---|---|
| 0 (uncut) | 23.56 | 12.17 | 0.77 | 0.50 |
| 8 | 60.16 | 18.26 | 0.95 | 0.23 |
| 16 | 75.68 | 21.33 | 0.97 | 0.16 |
| 32 | 95.50 | 25.88 | 0.98 | 0.10 |
| 48 | 100.03 | 28.26 | 0.97 | 0.12 |

(b) High-pass filtered after boosting

Table 5: **Amplification of low-frequency components is crucial to the success of Boost-and-Skip.** The performance values were evaluated on CelebA, with low-pass or high-pass filters applied after the boosted initialization. $f_{\text{cutoff, LPF}}$ represents the cut-off frequency for the low-pass filter, while $f_{\text{cutoff, HPF}}$ denotes the cut-off frequency for the high-pass filter. While the low-pass filtered cases maintain performance gains robustly across varying cut-off frequencies, high-pass filtering leads to immediate degradation, as seen in a significant performance drop starting at $f_{\text{cutoff, HPF}} = 8$. This suggests that the enhanced minority generation is primarily driven by the amplification of low-frequency components.

One may wonder: what if we employ $\gamma < 1.0$, effectively reversing the direction of minority generation? We found that this oppositional choice often exerts an interesting *quality-enhancing* effect on the generative process. Intuitively, initializing with $\gamma < 1.0$ can be interpreted as starting generation from a *high-density* region, which may guide the sampling process toward higher-quality outputs. Our empirical findings confirm this intuition.

Table 6 presents the effect of $\gamma$ on various quality and diversity metrics for CelebA. Note that these metrics were computed using the test dataset, rather than minority data as was employed for Table 1. This allows us to examine the quality-diversity tradeoff from a broader perspective, specifically within the ground-truth data manifold of CelebA. As shown in Table 6, using $\gamma < 1.0$ improves sample quality at the expense of diversity, demonstrating that Boost-and-Skip serves as a control mechanism not only for diversity but also for quality.

| $\gamma^2$ | Prec ↑ | Rec ↑ | Den ↑ | Cov ↑ |
|---|---|---|---|---|
| 1.0 | 0.851 | **0.627** | 1.290 | **0.940** |
| 0.8 | 0.874 | 0.588 | 1.384 | 0.938 |
| 0.6 | **0.881** | 0.586 | **1.495** | 0.936 |
| 0.5 | 0.879 | 0.565 | 1.478 | 0.926 |

Table 6: **Quality-enhancing effect of Boost-and-Skip on CelebA.** "Den" and "Cov" denote Density and Coverage (Naeem et al., 2020), an additional set of quality-diversity metrics. For computing these metrics, we used the CelebA test set as the baseline real data (rather than employing minority data as in Table 1) to assess the quality-diversity tradeoff within the ground-truth data manifold of CelebA.

Another key observation is that our approach is complementary to existing quality-enhancing techniques, such as Classifier-Free Guidance (CFG) (Ho & Salimans, 2022). This suggests that Boost-and-Skip can be integrated with such methods to further refine generative performance, offering additional flexibility in balancing quality and diversity.

### A.6. Discussion: Implications of Boost-and-Skip

We note that our framework has several important implications. The first one is that it improves the practical relevance of minority samplers by significantly reducing the computational overhead associated with existing methods. Secondly, our approach provides a simple mechanism for improving the diversity of diffusion models, a feature that has been largely absent in the research community. Another important point is to demonstrate that the pathological non-zero terminal SNR, arising from the training-inference mismatch (Lin et al., 2024; Hu et al., 2024), can actually be advantageous. Lastly, we highlight potential opportunities in the initialization of stochastic sampling, a largely under-explored area compared to deterministic samples of diffusion models.

## B. Proofs

### B.1. Proof of Proposition 3.1

**Proposition 3.1.** *Consider the temperature-scaled reverse VP-SDE in Eq. (9). Assuming the score function is optimal,* i.e., $s_{\boldsymbol{\theta}}(\boldsymbol{x}, t) = \nabla_{\boldsymbol{x}} \log p_t(\boldsymbol{x})$, *the marginal densities of samples generated by this SDE are not equal to* $\{\frac{1}{Z_t} p_t(\boldsymbol{x})^{1/\tau}\}_{t=0}^{T}$ *in general.*

*Proof.* We first rewrite Eq. (9) using the ground-truth score function $\nabla_{\boldsymbol{x}} \log p_t(\boldsymbol{x})$ under the optimality assumption:

$$\mathrm{d}\boldsymbol{x} = \left[ -\frac{1}{2}\beta(t)\boldsymbol{x} - \beta(t)\nabla_{\boldsymbol{x}} \log p_t(\boldsymbol{x})^{1/\tau} \right] \mathrm{d}t + \sqrt{\beta(t)}\mathrm{d}\tilde{\boldsymbol{w}}. \tag{26}$$

Let us assume that the SDE in Eq. (26) produces samples along $\{\frac{1}{Z_t}p_t(\boldsymbol{x})^{1/\tau}\}_{t=0}^T$. Then, we should have:

$$\frac{1}{Z_t}p_t(\boldsymbol{x})^{1/\tau} = \int p_{0t}(\boldsymbol{x}|\boldsymbol{x}_0)\frac{1}{Z}p_0(\boldsymbol{x}_0)^{1/\tau} \, \mathrm{d}\boldsymbol{x}_0, \tag{27}$$

where $Z$ represents a normalization constant w.r.t. $p_0$. We disprove this by providing a counterexample. Consider a dirac-delta density $p_0$ and a VP-SDE forward kernel $p_{0t}$:

$$p_0(\boldsymbol{x}_0) := \delta(\boldsymbol{x}_0 - \boldsymbol{\mu}), \tag{28}$$

$$p_{0t}(\boldsymbol{x}|\boldsymbol{x}_0) := \left( \frac{1}{\sqrt{2\pi(1-\alpha(t))}} \right)^d \exp\left( -\frac{\|\boldsymbol{x} - \sqrt{\alpha(t)}\boldsymbol{x}_0\|_2^2}{2(1-\alpha(t))} \right), \tag{29}$$

where $\alpha(t)$ represents the noise schedule, which is a function of $\beta(t)$. With this instantiation, we expand $p_t(\boldsymbol{x})^{1/\tau}$ on the LHS of Eq. (27) by using the definition of $p_t$:

$$p_t(\boldsymbol{x})^{1/\tau} = \left[ \int p_{0t}(\boldsymbol{x}|\boldsymbol{x}_0)p_0(\boldsymbol{x}_0) \, \mathrm{d}\boldsymbol{x}_0 \right]^{1/\tau} \tag{30}$$

$$= \left[ \int \left( \frac{1}{\sqrt{2\pi(1-\alpha(t))}} \right)^d \exp\left( -\frac{\|\boldsymbol{x} - \sqrt{\alpha(t)}\boldsymbol{x}_0\|_2^2}{2(1-\alpha(t))} \right) \delta(\boldsymbol{x}_0 - \boldsymbol{\mu}) \, \mathrm{d}\boldsymbol{x}_0 \right]^{1/\tau} \tag{31}$$

$$= \left( \frac{1}{\sqrt{2\pi(1-\alpha(t))}} \right)^{d/\tau} \exp\left( -\frac{\|\boldsymbol{x} - \sqrt{\alpha(t)}\boldsymbol{\mu}\|_2^2}{2\tau(1-\alpha(t))} \right). \tag{32}$$

Since $\frac{1}{Z_t}p_t(\boldsymbol{x})^{1/\tau}$ is Gaussian, we can determine the normalization constant $Z_t$, yielding the following expression for the LHS of Eq. (27):

$$\frac{1}{Z_t}p_t(\boldsymbol{x})^{1/\tau} = \left( \frac{1}{\sqrt{2\pi\tau(1-\alpha(t))}} \right)^d \exp\left( -\frac{\|\boldsymbol{x} - \sqrt{\alpha(t)}\boldsymbol{\mu}\|_2^2}{2\tau(1-\alpha(t))} \right). \tag{33}$$

On the other hand, the RHS of Eq. (27) is:

$$\int p_t(\boldsymbol{x}|\boldsymbol{x}_0)\frac{1}{Z}p_0(\boldsymbol{x}_0)^{1/\tau} \, \mathrm{d}\boldsymbol{x}_0 = \int \left( \frac{1}{\sqrt{2\pi(1-\alpha(t))}} \right)^d \exp\left( -\frac{\|\boldsymbol{x} - \sqrt{\alpha(t)}\boldsymbol{x}_0\|_2^2}{2(1-\alpha(t))} \right) \frac{1}{Z}\delta(\boldsymbol{x}_0 - \boldsymbol{\mu})^{1/\tau} \, \mathrm{d}\boldsymbol{x}_0 \tag{34}$$

$$= \left( \frac{1}{\sqrt{2\pi(1-\alpha(t))}} \right)^d \exp\left( -\frac{\|\boldsymbol{x} - \sqrt{\alpha(t)}\boldsymbol{\mu}\|_2^2}{2(1-\alpha(t))} \right), \tag{35}$$

which is not equal to the LHS (*i.e.*, Eq. (32)), *e.g.*, if $\tau \neq 1$, contradicting our initial assumption. This completes the proof.

$\square$

### B.2. Proof of Proposition 3.2

**Lemma B.1** (Solution to the VP-SDE). *Let $\boldsymbol{x}_0 \sim \mathcal{N}(\boldsymbol{\mu}_0, \boldsymbol{\Sigma}_0)$, and suppose $\boldsymbol{x}_t$ on $t \in (0, \infty)$ evolves according to*

$$d\boldsymbol{x} = -\frac{1}{2}\beta(t)\boldsymbol{x}dt + \sqrt{\beta(t)}d\boldsymbol{w} \tag{36}$$

*where $\beta(t)$ is a positive function which integrates to $\infty$. The forward process is Gaussian with mean and covariance*

$$\boldsymbol{\mu}_t := \alpha(t)\boldsymbol{\mu}_0, \tag{37}$$

$$\boldsymbol{\Sigma}_t := \boldsymbol{I} + \alpha(t)^2(\boldsymbol{\Sigma}_0 - \boldsymbol{I}), \tag{38}$$

*where $\alpha(t) = e^{-\frac{1}{2}\int_0^t \beta(s)\,ds}$ such that $\lim_{t\to\infty} \alpha(t) = 0$.*

*Proof.* Mean $\boldsymbol{\mu}_t$ of $\boldsymbol{x}_t$ is governed by the ODE

$$d\boldsymbol{\mu}_t/dt = \mathbb{E}[-\tfrac{1}{2}\beta(t)\boldsymbol{x}_t] = -\tfrac{1}{2}\beta(t)\boldsymbol{\mu}_t \tag{39}$$

so its solution is given as

$$\boldsymbol{\mu}_t = \alpha(t)\boldsymbol{\mu}_0. \tag{40}$$

The covariance $\boldsymbol{\Sigma}_t$ of $\boldsymbol{x}_t$ is governed by the ODE

$$d\boldsymbol{\Sigma}_t/dt = \mathbb{E}[-\tfrac{1}{2}\beta(t)\boldsymbol{x}_t(\boldsymbol{x}_t - \boldsymbol{\mu}_t)^\top] + \mathbb{E}[-\tfrac{1}{2}\beta(t)(\boldsymbol{x}_t - \boldsymbol{\mu}_t)\boldsymbol{x}_t^\top] + \mathbb{E}[\beta(t)] \tag{41}$$

$$= \mathbb{E}[-\tfrac{1}{2}\beta(t)(\boldsymbol{x}_t - \boldsymbol{\mu}_t)(\boldsymbol{x}_t - \boldsymbol{\mu}_t)^\top] + \mathbb{E}[-\tfrac{1}{2}\beta(t)(\boldsymbol{x}_t - \boldsymbol{\mu}_t)(\boldsymbol{x}_t - \boldsymbol{\mu}_t)^\top] + \beta(t) \tag{42}$$

$$\quad + \mathbb{E}[-\tfrac{1}{2}\beta(t)\boldsymbol{\mu}_t(\boldsymbol{x}_t - \boldsymbol{\mu}_t)^\top] + \mathbb{E}[-\tfrac{1}{2}\beta(t)(\boldsymbol{x}_t - \boldsymbol{\mu}_t)\boldsymbol{\mu}_t^\top] \tag{43}$$

$$= \mathbb{E}[-\tfrac{1}{2}\beta(t)(\boldsymbol{x}_t - \boldsymbol{\mu}_t)(\boldsymbol{x}_t - \boldsymbol{\mu}_t)^\top] + \mathbb{E}[-\tfrac{1}{2}\beta(t)(\boldsymbol{x}_t - \boldsymbol{\mu}_t)(\boldsymbol{x}_t - \boldsymbol{\mu}_t)^\top] + \beta(t) \tag{44}$$

$$= -\beta(t)(\boldsymbol{\Sigma}_t - \boldsymbol{I}) \tag{45}$$

whose solution is given as (solve the ODE after making the change of variables $\widehat{\boldsymbol{\Sigma}}_t = \boldsymbol{\Sigma}_t - \boldsymbol{I}$)

$$\boldsymbol{\Sigma}_t = \boldsymbol{I} + \alpha(t)^2(\boldsymbol{\Sigma}_0 - \boldsymbol{I}). \tag{46}$$

Since the initial distribution at $t = 0$ is a Gaussian and VP-SDE is a linear SDE, its solution is also a Gaussian. □

**Proposition 3.2.** *Let the data distribution be $\boldsymbol{x}(0) \sim \mathcal{N}(\boldsymbol{\mu}_0, \boldsymbol{\Sigma}_0)$, and assume the optimal score function $\boldsymbol{s}_{\boldsymbol{\theta}}(\boldsymbol{x}, t) = \nabla_{\boldsymbol{x}} \log p_t(\boldsymbol{x})$ trained on $p_0$ via the forward SDE in Eq. ([3](#)). Suppose the reverse SDE in Eq. ([10](#)) is initialized with $\hat{\boldsymbol{x}}(T_{skip}) \sim \mathcal{N}(\hat{\boldsymbol{\mu}}_{T_{skip}}, \hat{\boldsymbol{\Sigma}}_{T_{skip}})$. Then, the resulting generated distribution corresponds to $\hat{\boldsymbol{x}}(0) \sim \mathcal{N}(\hat{\boldsymbol{\mu}}_0, \hat{\boldsymbol{\Sigma}}_0)$, where*

$$\hat{\boldsymbol{\mu}}_0 := \boldsymbol{\mu}_0 + \alpha(T_{skip})\boldsymbol{\Sigma}_0\boldsymbol{\Sigma}_{T_{skip}}^{-1}(\hat{\boldsymbol{\mu}}_{T_{skip}} - \boldsymbol{\mu}_{T_{skip}}), \tag{47}$$

$$\hat{\boldsymbol{\Sigma}}_0 := \boldsymbol{\Sigma}_0 + \alpha(T_{skip})^2\boldsymbol{\Sigma}_0^2\boldsymbol{\Sigma}_{T_{skip}}^{-2}(\hat{\boldsymbol{\Sigma}}_{T_{skip}} - \boldsymbol{\Sigma}_{T_{skip}}). \tag{48}$$

*Here, $\boldsymbol{\mu}_{T_{skip}}$ and $\boldsymbol{\Sigma}_{T_{skip}}$ are defined as:*

$$\boldsymbol{\mu}_{T_{skip}} := \alpha(T_{skip})\boldsymbol{\mu}_0, \tag{49}$$

$$\boldsymbol{\Sigma}_{T_{skip}} := \boldsymbol{I} + \alpha(T_{skip})^2(\boldsymbol{\Sigma}_0 - \boldsymbol{I}). \tag{50}$$

*Proof.* By Lemma [B.1](#), the score function is given as

$$\nabla_{\boldsymbol{x}} \log p_t(\boldsymbol{x}) = \nabla_{\boldsymbol{x}} \log \mathcal{N}(\boldsymbol{x}|\boldsymbol{\mu}_t, \boldsymbol{\Sigma}_t) = -\boldsymbol{\Sigma}_t^{-1}(\boldsymbol{x} - \boldsymbol{\mu}_t) \tag{51}$$

such that the reverse-SDE is

$$d\hat{\boldsymbol{x}}_t = [-\tfrac{1}{2}\beta(t)\hat{\boldsymbol{x}}_t - \beta(t)\nabla \log p_t(\hat{\boldsymbol{x}}_t)]\,dt + \sqrt{\beta(t)}\,d\bar{\boldsymbol{w}}_t \tag{52}$$

$$= [-\tfrac{1}{2}\beta(t)\hat{\boldsymbol{x}}_t + \beta(t)\boldsymbol{\Sigma}_t^{-1}(\hat{\boldsymbol{x}}_t - \boldsymbol{\mu}_t)]\,dt + \sqrt{\beta(t)}\,d\bar{\boldsymbol{w}}_t \tag{53}$$

$$= \tfrac{1}{2}\beta(t)(2\boldsymbol{\Sigma}_t^{-1} - \boldsymbol{I})\hat{\boldsymbol{x}}_t\,dt - \beta(t)\boldsymbol{\Sigma}_t^{-1}\boldsymbol{\mu}_t\,dt + \sqrt{\beta(t)}\,d\hat{\boldsymbol{w}}_t \tag{54}$$

where $\hat{\boldsymbol{w}}_t$ is a reverse-time Brownian motion. This is also a linear SDE, and since $\hat{\boldsymbol{x}}_{T_{skip}}$ is assumed to be Gaussian, $\hat{\boldsymbol{x}}_0$ is also Gaussian. Hence, it suffices to derive the mean and covariance of $\hat{\boldsymbol{x}}_0$. Fix some $T > 0$ and make the change of variables $(\hat{\boldsymbol{y}}_s, s) \leftarrow (\hat{\boldsymbol{x}}_{T-t}, T - t)$ such that solving the SDE

$$d\hat{\boldsymbol{y}}_s = -\tfrac{1}{2}\beta(T - s)(2\boldsymbol{\Sigma}_{T-s}^{-1} - \boldsymbol{I})\hat{\boldsymbol{y}}_s\,ds + \beta(T - s)\boldsymbol{\Sigma}_{T-s}^{-1}\boldsymbol{\mu}_{T-s}\,ds + \sqrt{\beta(T - s)}\,d\boldsymbol{w}_s \tag{55}$$

from $s = 0$ to $T$ is equivalent to solving the original reverse-SDE Eq. (54) from $t = T$ to 0. We shall now derive the mean and covariance of $\hat{\boldsymbol{y}}_s$ at $s = T$. To this end, define the eigen-decomposition

$$\boldsymbol{\Sigma}_0 = \boldsymbol{Q}\boldsymbol{\Lambda}\boldsymbol{Q}^\top, \quad \boldsymbol{\Lambda} = \mathrm{diag}(\lambda_n) \tag{56}$$

and recall that $\boldsymbol{\mu}_t = \alpha(t)\boldsymbol{\mu}_0$ and $\boldsymbol{\Sigma}_t = \boldsymbol{I} + \alpha(t)^2(\boldsymbol{\Sigma}_0 - \boldsymbol{I})$.

The mean $\hat{\boldsymbol{\mu}}_s$ of $\hat{\boldsymbol{y}}_s$ is governed by the ODE

$$d\hat{\boldsymbol{\mu}}_s/ds = \mathbb{E}[-\tfrac{1}{2}\beta(T-s)(2\boldsymbol{\Sigma}_{T-s}^{-1} - \boldsymbol{I})\hat{\boldsymbol{y}}_s + \beta(T-s)\boldsymbol{\Sigma}_{T-s}^{-1}\boldsymbol{\mu}_{T-s}] \tag{57}$$

$$= -\tfrac{1}{2}\beta(T-s)(2\boldsymbol{\Sigma}_{T-s}^{-1} - \boldsymbol{I})\hat{\boldsymbol{\mu}}_s + \beta(T-s)\boldsymbol{\Sigma}_{T-s}^{-1}\boldsymbol{\mu}_{T-s} \tag{58}$$

$$= -\tfrac{1}{2}\alpha(T-s)\beta(T-s)\boldsymbol{Q}\,\mathrm{diag}(\tfrac{\alpha(T-s)^{-1} - \alpha(T-s)(\lambda_n - 1)}{1 + \alpha(T-s)^2(\lambda_n - 1)})\boldsymbol{Q}^\top\widehat{\boldsymbol{\mu}}_s \tag{59}$$

$$+ \alpha(T-s)\beta(T-s)\boldsymbol{Q}\,\mathrm{diag}(\tfrac{1}{1 + \alpha(T-s)^2(\lambda_n - 1)})\boldsymbol{Q}^\top\boldsymbol{\mu}_0. \tag{60}$$

Define $\phi(x; a) := \log(ax^2 + 1) - \log(x)$ such that

$$\frac{d\phi(\alpha(T-s);\lambda_n - 1)}{ds} = \frac{d\phi(\alpha(T-s);\lambda_n - 1)}{d\alpha(T-s)} \cdot \frac{d\alpha(T-s)}{d(T-s)} \cdot \frac{d(T-s)}{ds} \tag{61}$$

$$= -\frac{\alpha(T-s)^{-1} - \alpha(T-s)(\lambda_n - 1)}{1 + \alpha(T-s)^2(\lambda_n - 1)} \cdot \left(-\tfrac{1}{2}\beta(T-s)\alpha(T-s)\right) \cdot (-1) \tag{62}$$

$$= -\tfrac{1}{2}\beta(T-s)\alpha(T-s)\frac{\alpha(T-s)^{-1} - \alpha(T-s)(\lambda_n - 1)}{1 + \alpha(T-s)^2(\lambda_n - 1)}. \tag{63}$$

This motivates the transition matrix

$$\boldsymbol{\Psi}(s, s_0) := \boldsymbol{Q}\,\mathrm{diag}(e^{\phi(\alpha(T-s);\lambda_n - 1) - \phi(\alpha(T-s_0);\lambda_n - 1)})\boldsymbol{Q}^\top \tag{64}$$

$$= \boldsymbol{Q}\,\mathrm{diag}(\tfrac{(\lambda_n - 1)\alpha(T-s)^2 + 1}{\alpha(T-s)} \cdot \tfrac{\alpha(T-s_0)}{(\lambda_n - 1)\alpha(T-s_0)^2 + 1})\boldsymbol{Q}^\top \tag{65}$$

$$= \tfrac{\alpha(T-s_0)}{\alpha(T-s)}\boldsymbol{\Sigma}_{T-s}\boldsymbol{\Sigma}_{T-s_0}^{-1} \tag{66}$$

with which we can calculate the solution to Eq. (60) as

$$\hat{\boldsymbol{\mu}}_s = \boldsymbol{\Psi}(s, 0)\hat{\boldsymbol{\mu}}_0 + \int_0^s \boldsymbol{\Psi}(s, \sigma)\alpha(T-\sigma)\beta(T-\sigma)\boldsymbol{Q}\,\mathrm{diag}(\tfrac{1}{1 + \alpha(T-\sigma)^2(\lambda_n - 1)})\boldsymbol{Q}^\top\boldsymbol{\mu}_0\,d\sigma \tag{67}$$

$$= \tfrac{\alpha(T)}{\alpha(T-s)}\boldsymbol{\Sigma}_{T-s}\boldsymbol{\Sigma}_T^{-1}\widehat{\boldsymbol{\mu}}_0 + \boldsymbol{\mu}_{T-s} - \tfrac{\alpha(T)}{\alpha(T-s)}\boldsymbol{\Sigma}_{T-s}\boldsymbol{\Sigma}_T^{-1}\boldsymbol{\mu}_T \tag{68}$$

$$= \boldsymbol{\mu}_{T-s} + \tfrac{\alpha(T)}{\alpha(T-s)}\boldsymbol{\Sigma}_{T-s}\boldsymbol{\Sigma}_T^{-1}(\hat{\boldsymbol{\mu}}_0 - \boldsymbol{\mu}_T) \tag{69}$$

where the integral in the first line is calculated as

$$\int_0^s \boldsymbol{\Psi}(s, \sigma)\alpha(T-\sigma)\beta(T-\sigma)\boldsymbol{Q}\,\mathrm{diag}(\tfrac{1}{1 + \alpha(T-\sigma)^2(\lambda_n - 1)})\boldsymbol{Q}^\top\boldsymbol{\mu}_0\,d\sigma \tag{70}$$

$$= \int_0^s \alpha(T-\sigma)\beta(T-\sigma)\boldsymbol{Q}\,\mathrm{diag}(\tfrac{\alpha(T-\sigma)}{\alpha(T-s)} \cdot \tfrac{(\lambda_n - 1)\alpha(T-s)^2 + 1}{(\lambda_n - 1)\alpha(T-\sigma)^2 + 1})\,\mathrm{diag}(\tfrac{1}{1 + \alpha(T-\sigma)^2(\lambda_n - 1)})\boldsymbol{Q}^\top\boldsymbol{\mu}_0\,d\sigma \tag{71}$$

$$= \int_0^s \boldsymbol{Q}\,\mathrm{diag}(\tfrac{(\lambda_n - 1)\alpha(T-s)^2 + 1}{\alpha(T-s)})\,\mathrm{diag}(\tfrac{\alpha(T-\sigma)^2\beta(T-\sigma)}{(1 + \alpha(T-\sigma)^2(\lambda_n - 1))^2})\boldsymbol{Q}^\top\boldsymbol{\mu}_0\,d\sigma \tag{72}$$

$$= \boldsymbol{Q}\,\mathrm{diag}(\tfrac{(\lambda_n - 1)\alpha(T-s)^2 + 1}{\alpha(T-s)})\,\mathrm{diag}(2\int_0^s \tfrac{1}{2}\alpha(T-\sigma)\beta(T-\sigma)\tfrac{\alpha(T-\sigma)}{(1 + \alpha(T-\sigma)^2(\lambda_n - 1))^2}\,d\sigma)\boldsymbol{Q}^\top\boldsymbol{\mu}_0 \tag{73}$$

$$= \boldsymbol{Q}\,\mathrm{diag}(\tfrac{(\lambda_n - 1)\alpha(T-s)^2 + 1}{\alpha(T-s)})\,\mathrm{diag}(2\left[\tfrac{\alpha(T-\sigma)^2}{2((\lambda_n - 1)\alpha(T-\sigma)^2 + 1)}\right]_0^s)\boldsymbol{Q}^\top\boldsymbol{\mu}_0 \tag{74}$$

$$= \boldsymbol{Q}\,\mathrm{diag}(\tfrac{(\lambda_n - 1)\alpha(T-s)^2 + 1}{\alpha(T-s)})\,\mathrm{diag}(\left[\tfrac{\alpha(T-s)^2}{(\lambda_n - 1)\alpha(T-s)^2 + 1} - \tfrac{\alpha(T)^2}{(\lambda_n - 1)\alpha(T)^2 + 1}\right])\boldsymbol{Q}^\top\boldsymbol{\mu}_0 \tag{75}$$

$$= \boldsymbol{Q}\,\mathrm{diag}(\left[\alpha(T-s) - \tfrac{\alpha(T)^2}{\alpha(T-s)} \cdot \tfrac{(\lambda_n - 1)\alpha(T-s)^2 + 1}{(\lambda_n - 1)\alpha(T)^2 + 1}\right])\boldsymbol{Q}^\top\boldsymbol{\mu}_0 \tag{76}$$

$$= \alpha(T-s)\boldsymbol{\mu}_0 - \tfrac{\alpha(T)}{\alpha(T-s)} \cdot \boldsymbol{Q}\,\mathrm{diag}(\tfrac{(\lambda_n - 1)\alpha(T-s)^2 + 1}{(\lambda_n - 1)\alpha(T)^2 + 1})\boldsymbol{Q}^\top(\alpha(T)\boldsymbol{\mu}_0) \tag{77}$$

$$= \boldsymbol{\mu}_{T-s} - \tfrac{\alpha(T)}{\alpha(T-s)}\boldsymbol{\Sigma}_{T-s}\boldsymbol{\Sigma}_T^{-1}\boldsymbol{\mu}_T. \tag{78}$$

Setting $s = T = T_{\mathrm{skip}}$ in Eq. (69), we see that

$$\mathbb{E}[\hat{\boldsymbol{x}}_0] = \mathbb{E}[\hat{\boldsymbol{y}}_{T_{\mathrm{skip}}}] = \boldsymbol{\mu}_{T_{\mathrm{skip}} - T_{\mathrm{skip}}} + \tfrac{\alpha(T_{\mathrm{skip}})}{\alpha(T_{\mathrm{skip}} - T_{\mathrm{skip}})}\boldsymbol{\Sigma}_{T_{\mathrm{skip}} - T_{\mathrm{skip}}}\boldsymbol{\Sigma}_{T_{\mathrm{skip}}}^{-1}(\mathbb{E}[\hat{\boldsymbol{y}}_0] - \boldsymbol{\mu}_{T_{\mathrm{skip}}}) \tag{79}$$

$$= \boldsymbol{\mu}_0 + \alpha(T_{\mathrm{skip}})\boldsymbol{\Sigma}_0\boldsymbol{\Sigma}_{T_{\mathrm{skip}}}^{-1}(\mathbb{E}[\hat{\boldsymbol{x}}_{T_{\mathrm{skip}}}] - \boldsymbol{\mu}_{T_{\mathrm{skip}}}) \tag{80}$$

where we have used the fact that $\alpha(0) = 1$.

The covariance $\hat{\Sigma}_s$ of $\hat{y}_s$ is governed by the ODE

$$d\hat{\Sigma}_s/ds = \mathbb{E}[-\tfrac{1}{2}\beta(T-s)(2\Sigma_{T-s}^{-1} - I)\hat{y}_s(\hat{y}_s - \hat{\mu}_s)^\top] \tag{81}$$

$$+ \mathbb{E}[-\tfrac{1}{2}\beta(T-s)(\hat{y}_s - \hat{\mu}_s)y_s^\top(2\Sigma_{T-s}^{-1} - I)] + \beta(T-s) \tag{82}$$

$$= \mathbb{E}[-\tfrac{1}{2}\beta(T-s)(2\Sigma_{T-s}^{-1} - I)(\hat{y}_s - \hat{\mu}_s)(\hat{y}_s - \hat{\mu}_s)^\top] \tag{83}$$

$$+ \mathbb{E}[-\tfrac{1}{2}\beta(T-s)(\hat{y}_s - \hat{\mu}_s)(\hat{y}_s - \hat{\mu}_s)^\top(2\Sigma_{T-s}^{-1} - I)] + \beta(T-s) \tag{84}$$

$$= \mathbb{E}[-\beta(T-s)(2\Sigma_{T-s}^{-1} - I)(\hat{y}_s - \hat{\mu}_s)(\hat{y}_s - \hat{\mu}_s)^\top] + \beta(T-s) \tag{85}$$

$$= -\beta(T-s)(2\Sigma_{T-s}^{-1} - I)\hat{\Sigma}_s + \beta(T-s) \tag{86}$$

$$= -\beta(T-s)Q \operatorname{diag}(\tfrac{1-\alpha(T-s)^2(\lambda_n-1)}{1+\alpha(T-s)^2(\lambda_n-1)})Q^\top\hat{\Sigma}_s + \beta(T-s). \tag{87}$$

This motivates a similar transition matrix (note that we multiplied 2 to $\phi$ now)

$$\Psi(s, s_0) := Q \operatorname{diag}(e^{2\phi(\alpha(T-s);\lambda_n-1)-2\phi(\alpha(T-s_0);\lambda_n-1)})Q^\top \tag{88}$$

$$= Q \operatorname{diag}(\tfrac{\alpha(T-s_0)^2}{\alpha(T-s)^2} \cdot \tfrac{((\lambda_n-1)\alpha(T-s)^2+1)^2}{((\lambda_n-1)\alpha(T-s_0)^2+1)^2})Q^\top \tag{89}$$

$$= \tfrac{\alpha(T-s_0)^2}{\alpha(T-s)^2}\Sigma_{T-s}^2\Sigma_{T-s_0}^{-2}. \tag{90}$$

which produces the solution

$$\hat{\Sigma}_s = \Psi(s,0)\hat{\Sigma}_0 + \int_0^s \Psi(s,\sigma)\beta(T-\sigma)\,d\sigma \tag{91}$$

$$= \tfrac{\alpha(T)^2}{\alpha(T-s)^2}\Sigma_{T-s}^2\Sigma_T^{-2}\hat{\Sigma}_0 + \Sigma_{T-s} - \tfrac{\alpha(T)^2}{\alpha(T-s)^2}\Sigma_{T-s}^2\Sigma_T^{-1} \tag{92}$$

$$= \Sigma_{T-s} + \tfrac{\alpha(T)^2}{\alpha(T-s)^2}\Sigma_{T-s}^2\Sigma_T^{-2}(\hat{\Sigma}_0 - \Sigma_T) \tag{93}$$

where the integral in the first line is calculated as

$$\int_0^s \Psi(s,\sigma)\beta(T-\sigma)\,d\sigma \tag{94}$$

$$= \int_0^s Q \operatorname{diag}(\tfrac{\alpha(T-\sigma)^2}{\alpha(T-s)^2} \cdot \tfrac{((\lambda_n-1)\alpha(T-s)^2+1)^2}{((\lambda_n-1)\alpha(T-\sigma)^2+1)^2})Q^\top\beta(T-\sigma)\,d\sigma \tag{95}$$

$$= Q \operatorname{diag}(\tfrac{((\lambda_n-1)\alpha(T-s)^2+1)^2}{\alpha(T-s)^2}) \operatorname{diag}(\int_0^s \tfrac{\beta(T-\sigma)\alpha(T-\sigma)^2}{((\lambda_n-1)\alpha(T-\sigma)^2+1)^2}\,d\sigma)Q^\top \tag{96}$$

$$= Q \operatorname{diag}(\tfrac{((\lambda_n-1)\alpha(T-s)^2+1)^2}{\alpha(T-s)^2}) \operatorname{diag}(\left[\tfrac{\alpha(T-s)^2}{(\lambda_n-1)\alpha(T-s)^2+1} - \tfrac{\alpha(T)^2}{(\lambda_n-1)\alpha(T)^2+1}\right])Q^\top \tag{97}$$

$$= \tfrac{1}{\alpha(T-s)^2}\Sigma_{T-s}^2(\alpha(T-s)^2\Sigma_{T-s}^{-1} - \alpha(T)^2\Sigma_T^{-1}) \tag{98}$$

$$= \Sigma_{T-s} - \tfrac{\alpha(T)^2}{\alpha(T-s)^2}\Sigma_{T-s}^2\Sigma_T^{-1}. \tag{99}$$

Setting $s = T = T_{\text{skip}}$ in Eq. (93), we see that

$$\operatorname{Cov}[\hat{x}_0] = \operatorname{Cov}[\hat{y}_{T_{\text{skip}}}] = \Sigma_{T_{\text{skip}}-T_{\text{skip}}} + \tfrac{\alpha(T_{\text{skip}})^2}{\alpha(T_{\text{skip}}-T_{\text{skip}})^2}\Sigma_{T_{\text{skip}}-T_{\text{skip}}}^2\Sigma_{T_{\text{skip}}}^{-2}(\operatorname{Cov}[\hat{y}_0] - \Sigma_{T_{\text{skip}}}) \tag{100}$$

$$= \Sigma_0 + \alpha(T_{\text{skip}})^2\Sigma_0^2\Sigma_{T_{\text{skip}}}^{-2}(\operatorname{Cov}[\hat{x}_{T_{\text{skip}}}] - \Sigma_{T_{\text{skip}}}). \tag{101}$$

This concludes the proof. $\square$

## B.3. Proof of Corollary 3.3

**Corollary 3.3.** *Suppose $\Sigma_0 = \sigma_0^2 I$ and $\hat{\Sigma}_{T_{\text{skip}}} = \gamma^2 I$, and define the quantity (if it exists)*

$$\kappa := \sqrt{(\gamma^2 - 1)/(\sigma_0^2 - 1)}.$$

*The variance-amplification effect of $\hat{\Sigma}_0$ occurs iff*

$$T_{skip} \in \begin{cases} \varnothing & \text{if } \gamma \leq 1 \leq \sigma_0, \\ (\alpha^{-1}(\kappa), \infty) & \text{if } 1 < \gamma, \sigma_0, \\ [0, \alpha^{-1}(\kappa)) & \text{if } 1 > \gamma, \sigma_0, \\ [0, \infty) & \text{if } \sigma_0 \leq 1 \leq \gamma \text{ and } (\sigma_0, \gamma) \neq \mathbf{1}, \end{cases}$$

*where we define $\alpha^{-1}(\kappa) := 0$ when $\kappa > 1$.*

*Proof.* Under our assumption, variance-amplification occurs iff

$$\gamma^2 - 1 > \alpha(T_{\text{skip}})^2(\sigma_0^2 - 1). \tag{102}$$

**Case $\gamma \leq 1 \leq \sigma_0$.** Suppose some value of $T_{\text{skip}}$ offers variance amplification. Then

$$0 \geq \gamma^2 - 1 > \alpha(T_{\text{skip}})^2(\sigma_0^2 - 1) \geq 0 \tag{103}$$

which is a contradiction.

**Case $1 < \gamma, \sigma_0$.** In this case, Eq. (102) is equivalent to

$$\alpha(T_{\text{skip}}) < \kappa \tag{104}$$

which is equivalent to $T_{\text{skip}} \in (\alpha^{-1}(\kappa), \infty)$ since $\alpha$ is a monotone decreasing function.

**Case $1 > \gamma, \sigma_0$.** In this case, Eq. (102) is equivalent to

$$\alpha(T_{\text{skip}}) > \kappa \tag{105}$$

which is equivalent to $T_{\text{skip}} \in [0, \alpha^{-1}(\kappa))$ since $\alpha$ is a monotone decreasing function.

**Case $\sigma_0 \leq 1 \leq \gamma$ and $(\sigma_0, \gamma) \neq \mathbf{1}$.** In the case $\sigma_0 < 1 \leq \gamma$,

$$\gamma^2 - 1 \geq 0 > \alpha(T_{\text{skip}})(\sigma_0^2 - 1) \tag{106}$$

for all values of $T_{\text{skip}}$. Likewise, if $\sigma_0 \leq 1 < \gamma$,

$$\gamma^2 - 1 > 0 \leq \alpha(T_{\text{skip}})(\sigma_0^2 - 1) \tag{107}$$

for all values of $T_{\text{skip}}$. $\qquad\square$

### B.4. Proof of Proposition 3.4

**Proposition 3.4.** *Consider an empirical version of the discrete VP-SDE in Eq. (8):*

$$\boldsymbol{x}_{i-1} = \frac{1}{\sqrt{\alpha_i}}\big\{\boldsymbol{x}_i + (1 - \alpha_i)s_{\boldsymbol{\theta}}(\boldsymbol{x}_i, i)\big\} + \sqrt{1 - \alpha_i}\boldsymbol{z},$$

*where $i \in \{1, \ldots, N\}$. Assume the optimal score function, i.e., $s_{\boldsymbol{\theta}}(\boldsymbol{x}, i) = \nabla_{\boldsymbol{x}} \log p_i(\boldsymbol{x})$, which is trained on an arbitrary data distribution $p_0$. Consider two sample trajectories $\{\boldsymbol{x}_i\}_{i=0}^{N}$ and $\{\hat{\boldsymbol{x}}_i\}_{i=0}^{N_{skip}}$ where $N_{skip} < N$, which are initialized with distinct distributions: $\boldsymbol{x}_N \sim \mathcal{N}(\mathbf{0}, \boldsymbol{I})$ and $\hat{\boldsymbol{x}}_{N_{skip}} \sim \mathcal{N}(\mathbf{0}, \gamma^2\boldsymbol{I})$. Assuming that $\{\boldsymbol{x}_i\}_{i=0}^{N}$ is a bounded process such that $\|\boldsymbol{x}_i\|_2 < B$ (as in Xu et al. (2023)), the expected error between samples from these two trajectories at step $i \in \{0, \ldots, N_{skip} - 1\}$ is given by:*

$$\mathbb{E}[\|\boldsymbol{x}_i - \hat{\boldsymbol{x}}_i\|_2^2] \leq \frac{2C}{1 - \lambda^2} + \lambda^{2(N_{skip} - i)}(B^2 + \gamma^2 d), \tag{108}$$

*where $\lambda$ denotes the contraction rate:*

$$\lambda := \max_{j \in \{i+1, \ldots, N_{skip}\}} \sqrt{\alpha_j}\left(\frac{1 - \bar{\alpha}_{j-1}}{1 - \bar{\alpha}_j}\right), \tag{109}$$

*and $C := d(1 - \bar{\alpha}_{N_{skip}})$.*

*Proof.* The proof is based on the contraction theorem for discrete stochastic processes (Pham, 2008). However, unlike the original setup considered in Pham (2008), our focused trajectories start from distinct initial distributions and terminal timesteps: $\boldsymbol{x}_N \sim \mathcal{N}(\boldsymbol{0}, \boldsymbol{I})$ and $\hat{\boldsymbol{x}}_{N_{\text{skip}}} \sim \mathcal{N}(\boldsymbol{0}, \gamma^2 \boldsymbol{I})$. To address this, we leave out $\{\boldsymbol{x}_i\}_{i=N_{\text{skip}}+1}^{N}$ and focus on errors between $\{\boldsymbol{x}_i\}_{i=0}^{N_{\text{skip}}}$ and $\{\hat{\boldsymbol{x}}_i\}_{i=0}^{N_{\text{skip}}}$.

Employing $N_{\text{skip}}$ as the terminal timestep, the discrete contraction theorem (*i.e.*, Theorem 1 in Pham (2008)) gives the following error bound between $\boldsymbol{x}_i$ and $\hat{\boldsymbol{x}}_i$ at time $i \in \{0, \ldots, N_{\text{skip}} - 1\}$:

$$\mathbb{E}[\|\boldsymbol{x}_i - \hat{\boldsymbol{x}}_i\|_2^2] \leq \frac{2C}{1 - \lambda^2} + \lambda^{2(N_{\text{skip}}-i)}\mathbb{E}[\|\boldsymbol{x}_{N_{\text{skip}}} - \hat{\boldsymbol{x}}_{N_{\text{skip}}}\|_2^2]. \tag{110}$$

where $C := d(1 - \bar{\alpha}_{N_{\text{skip}}})$. Here $\lambda$ indicates the contraction rate that determines the speed of error-decaying:

$$\lambda := \max_{j \in \{i+1, \ldots, N_{\text{skip}}\}} \sqrt{\alpha_j} \left( \frac{1 - \bar{\alpha}_{j-1}}{1 - \bar{\alpha}_j} \right), \tag{111}$$

Next, we upper-bound the terminal error $\mathbb{E}[\|\boldsymbol{x}_{N_{\text{skip}}} - \hat{\boldsymbol{x}}_{N_{\text{skip}}}\|_2^2]$ as:

$$\mathbb{E}[\|\boldsymbol{x}_{N_{\text{skip}}} - \hat{\boldsymbol{x}}_{N_{\text{skip}}}\|_2^2] = \mathbb{E}[\|\boldsymbol{x}_{N_{\text{skip}}}\|_2^2] + \mathbb{E}[\|\hat{\boldsymbol{x}}_{N_{\text{skip}}}\|_2^2] + 2\mathbb{E}[\boldsymbol{x}_{N_{\text{skip}}}^{\mathsf{T}} \hat{\boldsymbol{x}}_{N_{\text{skip}}}] \tag{112}$$

$$= \mathbb{E}[\|\boldsymbol{x}_{N_{\text{skip}}}\|_2^2] + \mathbb{E}[\|\hat{\boldsymbol{x}}_{N_{\text{skip}}}\|_2^2] \tag{113}$$

$$\leq B^2 + \gamma^2 d, \tag{114}$$

where the second equality is due to the independent initialization $\hat{\boldsymbol{x}}_{N_{\text{skip}}} \sim \mathcal{N}(\boldsymbol{0}, \gamma^2 \boldsymbol{I})$. The inequality comes from the bounded assumption $\|\boldsymbol{x}_i\|_2 < B$. Now plugging the inequality in Eq. (114) into Eq. (110) yields the desired result:

$$\mathbb{E}[\|\boldsymbol{x}_i - \hat{\boldsymbol{x}}_i\|_2^2] \leq \frac{2C}{1 - \lambda^2} + \lambda^{2(N_{\text{skip}}-i)}\mathbb{E}[\|\boldsymbol{x}_{N_{\text{skip}}} - \hat{\boldsymbol{x}}_{N_{\text{skip}}}\|_2^2] \tag{115}$$

$$\leq \frac{2C}{1 - \lambda^2} + \lambda^{2(N_{\text{skip}}-i)}(B^2 + \gamma^2 d). \tag{116}$$

$$\tag{117}$$

This completes the proof. $\qquad\square$

## B.5. Proof of Proposition A.1

**Proposition A.1.** *Consider the same data distribution $p_0$ and the optimal score function $\boldsymbol{s}_{\boldsymbol{\theta}}(\boldsymbol{x}, t)$ as Proposition 3.2. Suppose the PF-ODE in Eq. (19) is initialized with $\hat{\boldsymbol{x}}(T_{skip}) \sim \mathcal{N}(\hat{\boldsymbol{\mu}}_{T_{skip}}, \hat{\boldsymbol{\Sigma}}_{T_{skip}})$. Then, the resulting distribution at $t = 0$ is given by $\hat{\boldsymbol{x}}_{ODE}(0) \sim \mathcal{N}(\hat{\boldsymbol{\mu}}_{0,ODE}, \hat{\boldsymbol{\Sigma}}_{0,ODE})$, where*

$$\hat{\boldsymbol{\mu}}_{0,ODE} := \boldsymbol{\mu}_0 + \boldsymbol{\Sigma}_0^{1/2} \boldsymbol{\Sigma}_{T_{skip}}^{-1/2} (\hat{\boldsymbol{\mu}}_{T_{skip}} - \boldsymbol{\mu}_{T_{skip}}), \tag{118}$$

$$\hat{\boldsymbol{\Sigma}}_{0,ODE} := \boldsymbol{\Sigma}_0 \boldsymbol{\Sigma}_{T_{skip}}^{-1} \hat{\boldsymbol{\Sigma}}_{T_{skip}}. \tag{119}$$

*Here, $\boldsymbol{\mu}_{T_{skip}}$ and $\boldsymbol{\Sigma}_{T_{skip}}$ are defined as in Eqs. (15) and (16), respectively.*

*Proof.* By Lemma B.1, the score function is given as

$$\nabla_{\boldsymbol{x}} \log p_t(\boldsymbol{x}) = \nabla_{\boldsymbol{x}} \log \mathcal{N}(\boldsymbol{x} | \boldsymbol{\mu}_t, \boldsymbol{\Sigma}_t) = -\boldsymbol{\Sigma}_t^{-1}(\boldsymbol{x} - \boldsymbol{\mu}_t) \tag{120}$$

such that the PF-ODE is

$$d\hat{\boldsymbol{x}}_t = [-\tfrac{1}{2}\beta(t)\hat{\boldsymbol{x}}_t - \tfrac{1}{2}\beta(t)\nabla \log p_t(\hat{\boldsymbol{x}}_t)]\, dt \tag{121}$$

$$= [-\tfrac{1}{2}\beta(t)\hat{\boldsymbol{x}}_t + \tfrac{1}{2}\beta(t)\boldsymbol{\Sigma}_t^{-1}(\hat{\boldsymbol{x}}_t - \alpha(t)\boldsymbol{\mu}_0)]\, dt \tag{122}$$

$$= \tfrac{1}{2}\beta(t)(\boldsymbol{\Sigma}_t^{-1} - \boldsymbol{I})\hat{\boldsymbol{x}}_t\, dt - \tfrac{1}{2}\beta(t)\boldsymbol{\Sigma}_t^{-1}\boldsymbol{\mu}_t\, dt. \tag{123}$$

or with change of variables $(\hat{\boldsymbol{y}}_s, s) \leftarrow (\hat{\boldsymbol{x}}_{T-t}, T-t)$,

$$d\hat{\boldsymbol{y}}_s = -\tfrac{1}{2}\beta(T-s)(\boldsymbol{\Sigma}_{T-s}^{-1} - \boldsymbol{I})\hat{\boldsymbol{y}}_s \, ds + \tfrac{1}{2}\beta(T-s)\boldsymbol{\Sigma}_{T-s}^{-1}\boldsymbol{\mu}_{T-s} \, ds \tag{124}$$

$$= \tfrac{1}{2}\alpha(T-s)\beta(T-s)\boldsymbol{Q}\operatorname{diag}(\tfrac{\alpha(T-s)(\lambda_n-1)}{1+\alpha(T-s)^2(\lambda_n-1)})\boldsymbol{Q}^\top \hat{\boldsymbol{y}}_s \, ds \tag{125}$$

$$+ \tfrac{1}{2}\alpha(T-s)\beta(T-s)\boldsymbol{Q}\operatorname{diag}(\tfrac{1}{1+\alpha(T-s)^2(\lambda_n-1)})\boldsymbol{Q}^\top \boldsymbol{\mu}_0 \, ds. \tag{126}$$

Define

$$\phi(x; a) = \tfrac{1}{2}\log(ax^2 + 1) \tag{127}$$

such that

$$\tfrac{d\phi(\alpha(T-s);\lambda_n-1)}{ds} = \tfrac{d\phi(\alpha(T-s);\lambda_n-1)}{d\alpha(T-s)} \cdot \tfrac{d\alpha(T-s)}{d(T-s)} \cdot \tfrac{d(T-s)}{ds} \tag{128}$$

$$= \tfrac{\alpha(T-s)(\lambda_n-1)}{1+\alpha(T-s)^2(\lambda_n-1)} \cdot \left(-\tfrac{1}{2}\beta(T-s)\alpha(T-s)\right) \cdot (-1) \tag{129}$$

$$= \tfrac{1}{2}\alpha(T-s)\beta(T-s)\tfrac{\alpha(T-s)(\lambda_n-1)}{1+\alpha(T-s)^2(\lambda_n-1)}. \tag{130}$$

This motivates the transition matrix

$$\boldsymbol{\Psi}(s, s_0) := \boldsymbol{Q}\operatorname{diag}(e^{\phi(\alpha(T-s);\lambda_n-1)-\phi(\alpha(T-s_0);\lambda_n-1)})\boldsymbol{Q}^\top \tag{131}$$

$$= \boldsymbol{Q}\operatorname{diag}(\tfrac{\sqrt{(\lambda_n-1)\alpha(T-s)^2+1}}{\sqrt{(\lambda_n-1)\alpha(T-s_0)^2+1}})\boldsymbol{Q}^\top \tag{132}$$

$$= \boldsymbol{\Sigma}_{T-s}^{1/2}\boldsymbol{\Sigma}_{T-s_0}^{-1/2} \tag{133}$$

with which we can calculate the solution as

$$\hat{\boldsymbol{y}}_s = \boldsymbol{\Psi}(s,0)\hat{\boldsymbol{y}}_0 + \int_0^s \tfrac{1}{2}\boldsymbol{\Psi}(s,\sigma)\alpha(T-\sigma)\beta(T-\sigma)\boldsymbol{Q}\operatorname{diag}(\tfrac{1}{1+\alpha(T-\sigma)^2(\lambda_n-1)})\boldsymbol{Q}^\top \boldsymbol{\mu}_0 \, d\sigma \tag{134}$$

$$= \boldsymbol{\Sigma}_{T-s}^{1/2}\boldsymbol{\Sigma}_T^{-1/2}\hat{\boldsymbol{y}}_0 + \boldsymbol{\mu}_{T-s} - \boldsymbol{\Sigma}_{T-s}^{1/2}\boldsymbol{\Sigma}_T^{-1/2}\boldsymbol{\mu}_T \tag{135}$$

$$= \boldsymbol{\mu}_{T-s} + \boldsymbol{\Sigma}_{T-s}^{1/2}\boldsymbol{\Sigma}_T^{-1/2}(\hat{\boldsymbol{y}}_0 - \boldsymbol{\mu}_T) \tag{136}$$

where the integral in the first line is calculated as

$$\int_0^s \tfrac{1}{2}\boldsymbol{\Psi}(s,\sigma)\alpha(T-\sigma)\beta(T-\sigma)\boldsymbol{Q}\operatorname{diag}(\tfrac{1}{1+\alpha(T-\sigma)^2(\lambda_n-1)})\boldsymbol{Q}^\top \boldsymbol{\mu}_0 \, d\sigma \tag{137}$$

$$= \boldsymbol{Q}\operatorname{diag}(\sqrt{(\lambda_n-1)\alpha(T-s)^2+1})\operatorname{diag}(\int_0^s \tfrac{1}{2}\alpha(T-\sigma)\beta(T-\sigma)\tfrac{1}{(1+\alpha(T-\sigma)^2(\lambda_n-1))^{3/2}} \, d\sigma)\boldsymbol{Q}^\top \boldsymbol{\mu}_0 \tag{138}$$

$$= \boldsymbol{Q}\operatorname{diag}(\sqrt{(\lambda_n-1)\alpha(T-s)^2+1})\operatorname{diag}(\left[\tfrac{\alpha(T-\sigma)}{\sqrt{(\lambda_n-1)\alpha(T-\sigma)^2+1}}\right]_0^s)\boldsymbol{Q}^\top \boldsymbol{\mu}_0 \tag{139}$$

$$= \boldsymbol{Q}\operatorname{diag}(\sqrt{(\lambda_n-1)\alpha(T-s)^2+1})\operatorname{diag}(\left[\tfrac{\alpha(T-s)}{\sqrt{(\lambda_n-1)\alpha(T-s)^2+1}} - \tfrac{\alpha(T)}{\sqrt{(\lambda_n-1)\alpha(T)^2+1}}\right])\boldsymbol{Q}^\top \boldsymbol{\mu}_0 \tag{140}$$

$$= \alpha(T-s)\boldsymbol{\mu}_0 - \alpha(T)\boldsymbol{\Sigma}_{T-s}^{1/2}\boldsymbol{\Sigma}_T^{-1/2}\boldsymbol{\mu}_0 \tag{141}$$

$$= \boldsymbol{\mu}_{T-s} - \boldsymbol{\Sigma}_{T-s}^{1/2}\boldsymbol{\Sigma}_T^{-1/2}\boldsymbol{\mu}_T. \tag{142}$$

Setting $s = T = T_{\text{skip}}$ in Eq. (136),

$$\hat{\boldsymbol{x}}_0 = \hat{\boldsymbol{y}}_{T_{\text{skip}}} = \boldsymbol{\mu}_{T_{\text{skip}}-T_{\text{skip}}} + \boldsymbol{\Sigma}_{T_{\text{skip}}-T_{\text{skip}}}^{1/2}\boldsymbol{\Sigma}_{T_{\text{skip}}}^{-1/2}(\hat{\boldsymbol{y}}_0 - \boldsymbol{\mu}_{T_{\text{skip}}}) \tag{143}$$

$$= \boldsymbol{\mu}_0 + \boldsymbol{\Sigma}_0^{1/2}\boldsymbol{\Sigma}_{T_{\text{skip}}}^{-1/2}(\hat{\boldsymbol{x}}_{T_{\text{skip}}} - \boldsymbol{\mu}_{T_{\text{skip}}}) \tag{144}$$

$$= \boldsymbol{\mu}_0 + \boldsymbol{\Sigma}_0^{1/2}\boldsymbol{\Sigma}_{T_{\text{skip}}}^{-1/2}(\mathbb{E}[\hat{\boldsymbol{x}}_{T_{\text{skip}}}] - \boldsymbol{\mu}_{T_{\text{skip}}}) + \boldsymbol{\Sigma}_0^{1/2}\boldsymbol{\Sigma}_{T_{\text{skip}}}^{-1/2}(\hat{\boldsymbol{x}}_{T_{\text{skip}}} - \mathbb{E}[\hat{\boldsymbol{x}}_{T_{\text{skip}}}]). \tag{145}$$

It follows that if $\hat{\boldsymbol{x}}_{T_{\text{skip}}}$ is Gaussian, $\hat{\boldsymbol{x}}_0$ is also Gaussian with

$$\mathbb{E}[\hat{\boldsymbol{x}}_0] = \boldsymbol{\mu}_0 + \boldsymbol{\Sigma}_0^{1/2}\boldsymbol{\Sigma}_{T_{\text{skip}}}^{-1/2}(\mathbb{E}[\hat{\boldsymbol{x}}_{T_{\text{skip}}}] - \boldsymbol{\mu}_{T_{\text{skip}}}), \tag{146}$$

$$\operatorname{Cov}[\hat{\boldsymbol{x}}_0] = \boldsymbol{\Sigma}_0\boldsymbol{\Sigma}_{T_{\text{skip}}}^{-1}\operatorname{Cov}[\hat{\boldsymbol{x}}_{T_{\text{skip}}}]. \tag{147}$$

This concludes the proof. □

### B.6. Proof of Proposition A.2

**Proposition A.2.** *Assume the same setup as Theorem 3 in Xu et al. (2023), i.e., $\|t\nabla \log p_t(\boldsymbol{x})\| \le L_1$ and $\|\boldsymbol{x}_t\| < B/2$ for any $\boldsymbol{x}_t$ in the support of $p_t$ and reverse-SDE trajectories. Let*

$$\boldsymbol{x}_{t_{\min}}^{\mathrm{SDE}} = \mathrm{SDE}(\boldsymbol{x}_{t_{\max}}, t_{\max} \to t_{\min}), \ \boldsymbol{y}_{t_{\min}}^{\mathrm{SDE}} = \mathrm{SDE}(\boldsymbol{y}_{t_{\max}}, t_{\max} \to t_{\min}) \tag{148}$$

$$\boldsymbol{x}_{t_{\min}}^{\mathrm{ODE}} = \mathrm{ODE}(\boldsymbol{x}_{t_{\max}}, t_{\max} \to t_{\min}), \ \boldsymbol{y}_{t_{\min}}^{\mathrm{ODE}} = \mathrm{ODE}(\boldsymbol{y}_{t_{\max}}, t_{\max} \to t_{\min}) \tag{149}$$

*where* SDE *and* ODE *denote solutions to the reverse-SDE and PF-ODE, respectively. Then*

$$\mathrm{TV}(\boldsymbol{x}_{t_{\min}}^{\mathrm{ODE}}, \boldsymbol{y}_{t_{\min}}^{\mathrm{ODE}}) = \mathrm{TV}(\boldsymbol{x}_{t_{\max}}, \boldsymbol{y}_{t_{\max}}) \tag{150}$$

$$\mathrm{TV}(\boldsymbol{x}_{t_{\min}}^{\mathrm{SDE}}, \boldsymbol{y}_{t_{\min}}^{\mathrm{SDE}}) \le \left(1 - 2Q\left(\frac{B}{2\sqrt{\alpha(t_{\max})^{-2} - \alpha(t_{\min})^{-2}}}\right) \cdot e^{-BL_1/t_{\min} - L_1^2 \alpha(t_{\max})^{-2}/t_{\min}^2}\right) \mathrm{TV}(\boldsymbol{x}_{t_{\max}}, \boldsymbol{y}_{t_{\max}}) \tag{151}$$

*where $Q(r) := \mathbb{P}(\varepsilon \ge r)$ for $\varepsilon \sim \mathcal{N}(0, 1)$.*

*Proof.* The first equality follows by noting that the total variation distance is invariant under bijective maps due to the data processing inequality. We now prove the second inequality. Suppose $\boldsymbol{x}_t$ follows the reverse SDE, and consider the change of variables $\bar{\boldsymbol{x}}_t = \boldsymbol{x}_t/\alpha(t)$. Then, by the Itô formula,

$$d\bar{\boldsymbol{x}}_t = (-\tfrac{1}{2}\beta(t)\alpha(t))(-\alpha(t)^{-2})\boldsymbol{x}_t \, dt + \alpha(t)^{-1} d\boldsymbol{x}_t \tag{152}$$

$$= (-\tfrac{1}{2}\beta(t)\alpha(t))(-\alpha(t)^{-2})\boldsymbol{x}_t \, dt + \alpha(t)^{-1}([-\tfrac{1}{2}\beta(t)\boldsymbol{x}_t - \beta(t)\nabla \log p_t(\boldsymbol{x}_t)] \, dt + \sqrt{\beta(t)} \, d\bar{\boldsymbol{w}}_t) \tag{153}$$

$$= -\alpha(t)^{-1}\beta(t)\nabla \log p_t(\boldsymbol{x}_t) \, dt + \alpha(t)^{-1}\sqrt{\beta(t)} \, d\bar{\boldsymbol{w}}_t \tag{154}$$

$$= -\alpha(t)^{-2}\beta(t)\nabla \log \bar{p}_t(\bar{\boldsymbol{x}}_t) \, dt + \alpha(t)^{-1}\sqrt{\beta(t)} \, d\bar{\boldsymbol{w}}_t \tag{155}$$

$$= -(\alpha(t)^{-2})'\nabla \log \bar{p}_t(\bar{\boldsymbol{x}}_t) \, dt + \sqrt{(\alpha(t)^{-2})'} \, d\bar{\boldsymbol{w}}_t \tag{156}$$

where $\bar{p}_t$ is the density of $\boldsymbol{x}_t/\alpha(t)$ for $\boldsymbol{x}_t \sim p_t$, and we have used the change-of-variables formula of probability density functions at the fourth line. Next, using the change of time variable $\bar{t} = \alpha(t)^{-2}$, we get

$$d\bar{\boldsymbol{x}}_{\bar{t}} = -\nabla \log \bar{p}_{g^{-1}(\bar{t})}(\bar{\boldsymbol{x}}_{\bar{t}}) \, d\bar{t} + d\bar{\boldsymbol{w}}_{\bar{t}}. \tag{157}$$

Then following an analogous process as the proof of Lemma 5 in Xu et al. (2023), we obtain the inequality

$$\mathrm{TV}(\bar{\boldsymbol{x}}_{\bar{t}_{\min}}, \bar{\boldsymbol{y}}_{\bar{t}_{\min}}) \le \left(1 - 2Q\left(\frac{B}{2\sqrt{\bar{t}_{\max} - \bar{t}_{\min}}}\right) \cdot e^{-BL_1/t_{\min} - L_1^2 \bar{t}_{\max}/t_{\min}^2}\right) \mathrm{TV}(\bar{\boldsymbol{x}}_{\bar{t}_{\max}}, \bar{\boldsymbol{y}}_{\bar{t}_{\max}}) \tag{158}$$

where $\bar{t}_{\min} = \alpha(t_{\min})^{-2}, \bar{t}_{\max} = \alpha(t_{\max})^{-2}$. Since the map $\boldsymbol{y}_s \mapsto \boldsymbol{y}_{\alpha(s)^2}/\alpha(s)$ is bijective, we again have

$$\mathrm{TV}(\bar{\boldsymbol{x}}_{\bar{t}_{\min}}, \bar{\boldsymbol{y}}_{\bar{t}_{\min}}) = \mathrm{TV}(\boldsymbol{x}_{t_{\min}}, \boldsymbol{y}_{t_{\min}}), \tag{159}$$

$$\mathrm{TV}(\bar{\boldsymbol{x}}_{\bar{t}_{\max}}, \bar{\boldsymbol{y}}_{\bar{t}_{\max}}) = \mathrm{TV}(\boldsymbol{x}_{t_{\max}}, \boldsymbol{y}_{t_{\max}}), \tag{160}$$

by the data processing inequality. Plugging this relation into Eq. (158) yields the contraction result for the reverse SDE. □

## C. Implementation Details

**Pretrained models.** The pretrained model for CelebA was constructed by ourselves by following the settings in Um et al. (2023). The models for LSUN-Bedrooms and ImageNet were taken from the checkpoints given by Dhariwal & Nichol (2021). For the results on ImageNet $256 \times 256$. we respect the approaches in Sehwag et al. (2022); Um et al. (2023); Um & Ye (2024b) and leverage the upscaling model developed in Dhariwal & Nichol (2021).

**Baselines.** The BigGAN model is based upon the same architecture used in Choi et al. (2020)[3], and we follow the training setup outlined in the official project page of BigGAN[4]. The StyleGAN results were obtained via the pretrained model

---

[3]https://github.com/ermongroup/fairgen
[4]https://github.com/ajbrock/BigGAN-PyTorch

offered by Karras et al. (2019)[5]. The ADM (Dhariwal & Nichol, 2021) baselines used in our experiments employed the same pretrained models as those used in our framework, which is provided in the authors' codebase[6]. The results of LDM (Rombach et al., 2022) were obtained from the checkpoint offered by Rombach et al. (2022)[7]. For the EDM (Karras et al., 2022) baseline, we used the checkpoint given in the official project page of (Karras et al., 2022)[8]. The DiT (Peebles & Xie, 2022) baseline employed the pretrained model provided in the official code repository[9]. The authors of CADS (Sadat et al., 2023) do not provide the official codebase, so we implemented by ourselves based upon the pseudocode provided in the manuscript (Sadat et al., 2023).

The implementation of Sehwag et al. (2022) is based upon the descriptions provided in their original paper, employing the same pretrained models as ours (*i.e.*, the ADM checkpoints (Dhariwal & Nichol, 2021)). Specifically on CelebA, we created an out-of-distribution (OOD) classifier to distinguish whether an input belongs to CelebA or other datasets (*e.g.*, ImageNet). We then incorporated the negative log-likelihood gradient of the classifier (targeting the in-distribution class) into ancestral sampling to yield low-density guidance. For implementing Um et al. (2023), we followed the settings described in their paper for all datasets under consideration. Specifically, using the same ADM pretrained models as ours, we applied U-Net encoders for minority classifiers and utilized the entire training set to build the classifiers, except for the minority predictor on LSUN-Bedrooms, where only 10% of the training set was used. For the ImageNet $256 \times 256$ results of Um et al. (2023), we used the upscaling model (Dhariwal & Nichol, 2021) as described in Um et al. (2023).

In the additional CelebA baseline with classifier guidance targeting minority annotations (*i.e.*, ADM-ML in Table 1), we respected the same settings outlined in Um et al. (2023); Um & Ye (2024b). In particular, the classifier was trained to predict four minority attributes: (i) "Bald"; (ii) "Eyeglasses"; (iii) "Mustache"; (iv) "Wearing_Hat". During inference, we generated samples by combining random selections of these four attributes (*e.g.*, bald but not wearing glasses) using the classifier guidance. The backbone model for ADM-ML was the same as ours. The results of Um & Ye (2024b) is based upon their the official codebase[10], respecting their recommended settings reported in the original paper (Um & Ye, 2024b). The temperature sampling results were obtained from the method proposed in (Dhariwal & Nichol, 2021) (in Section G), *i.e.*, by using a $\tau$-scaled score model $s_{\theta}(x, t)/\tau$ with $\tau = 1.01$.

**Evaluation metrics.** To compute the Local Outlier Factor (LOF) (Breunig et al., 2000) for the generated samples, we used the PyOD implementation (Zhao et al., 2019)[11]. The number of nearest neighbors for calculating AvgkNN and LOF was set to 5 and 20, respectively, as these are commonly used values in practice. Following the approach in Sehwag et al. (2022); Um et al. (2023), both AvgkNN and LOF were computed in the feature space of ResNet-50. The Rarity Score (Han et al., 2022) was computed with $k = 5$ using the implementation available on the official project page[12].

Clean Fréchet Inception Distance (cFID) (Parmar et al., 2022) was evaluated using the official implementation[13]. Spatial FID (sFID) (Nash et al., 2021) was computed based on the official PyTorch FID code (Heusel et al., 2017)[14] with modifications to utilize spatial features (*i.e.*, the first 7 channels from the intermediate `mixed_6/conv` feature maps), rather than the standard `pool_3` inception features. The results for Improved Precision & Recall (Kynkäänniemi et al., 2019) were obtained with $k = 5$ using the official codebase from Han et al. (2022). To evaluate the proximity to low-likelihood instances at the data tail, we used the least probable instances as baseline real data for computing the quality metrics. Specifically for CelebA, we selected the 10,000 real samples with the highest AvgkNN values. For LSUN-Bedrooms and ImageNet, we used the most unique 50,000 samples, which had the highest AvgkNN values, as baseline real data. All quality and diversity metrics were computed using 30,000 generated samples.

**Hyperparameters.** Our hyperparameter selection $(\gamma, \Delta_t)$ followed a two-step approach: first, we determined an appropriate $\Delta_t$ that ensures a non-negligible $\alpha(T_{\text{skip}})$ (where $T_{\text{skip}} := T - \Delta_t$), and then we performed a grid search to select $\gamma$. We empirically found that our framework is not that sensitive to the choice of $\Delta_t$, and in practice, setting $\Delta_t$ such that $\alpha(T_{\text{skip}}) > 0.01$ generally yields strong performance on low-resolution datasets (*e.g.*, CelebA and ImageNet $64 \times 64$). For

---

[5] https://github.com/NVlabs/stylegan
[6] https://github.com/openai/guided-diffusion
[7] https://github.com/CompVis/latent-diffusion
[8] https://github.com/NVlabs/edm
[9] https://github.com/facebookresearch/DiT
[10] https://github.com/soobin-um/sg-minority
[11] https://pyod.readthedocs.io/en/latest/
[12] https://github.com/hichoe95/Rarity-Score
[13] https://github.com/GaParmar/clean-fid
[14] https://github.com/mseitzer/pytorch-fid

high-resolution benchmarks (*e.g.*, LSUN-Bedrooms), a lower threshold of $\alpha(T_{\text{skip}}) > 0.005$ was sufficient, as these datasets are more sensitive to noise intensity (Nichol & Dhariwal, 2021).

For CelebA, we conducted a grid search over $\gamma^2 = \{4.0, 8.0, 12.0, 16.0, 18.0, 20.0\}$, while for LSUN-Bedrooms, we searched over $\gamma^2 = \{2.0, 4.0, 6.0, 7.0, 7.5, 8.0\}$. For the ImageNet results, the search was performed over $\gamma^2 = \{2.0, 4.0, 6.0, 6.5, 7.0, 8.0\}$. Based on this, we selected the following final values: (i) $(\gamma^2, \Delta_t) = (18.0, 3)$ for CelebA; (ii) $(\gamma^2, \Delta_t) = (7.5, 0)$ for LSUN-Bedrooms; and (iii) $(\gamma^2, \Delta_t) = (6.5, 3)$ for the ImageNet cases.

We employed a global setting of 250 timesteps for sampling across all diffusion-based samplers, including both the baseline methods and our approach. For empirical analyses, such as the ablation studies, we reduced the number of timesteps to 100 for efficiency.

**Other details.** Our implementation is based on PyTorch (Paszke et al., 2019), and experiments were performed on twin NVIDIA A100 GPUs. Code is available at `https://github.com/soobin-um/BnS`.

## D. Additional Experimental Results

### D.1. Neighborhood distance distributions

Figure 7 illustrates neighborhood metric results compared on our interested four benchmarks. Observe that for all three metrics, our approach performs consistently well, rivaling to computationally intensive guided minority samplers like Um & Ye (2024b). This highlight the practical significance of Boost-and-Skip, which can achieve great performance improvements in promoting minority features with significantly less computations.

### D.2. Additional generated samples

To facilitate a more comprehensive qualitative comparison among the samplers, we present an extensive set of generated samples across all considered datasets. See Figures 8-11 for details. Observe that samples generated by our approach tend to capture unique dataset features, comparable to those produced by guided sampling techniques. This further highlights the key advantage of our method – achieving strong minority generation performance with significantly lower computational costs than existing guidance-based approaches (Sehwag et al., 2022; Um et al., 2023; Um & Ye, 2024b).

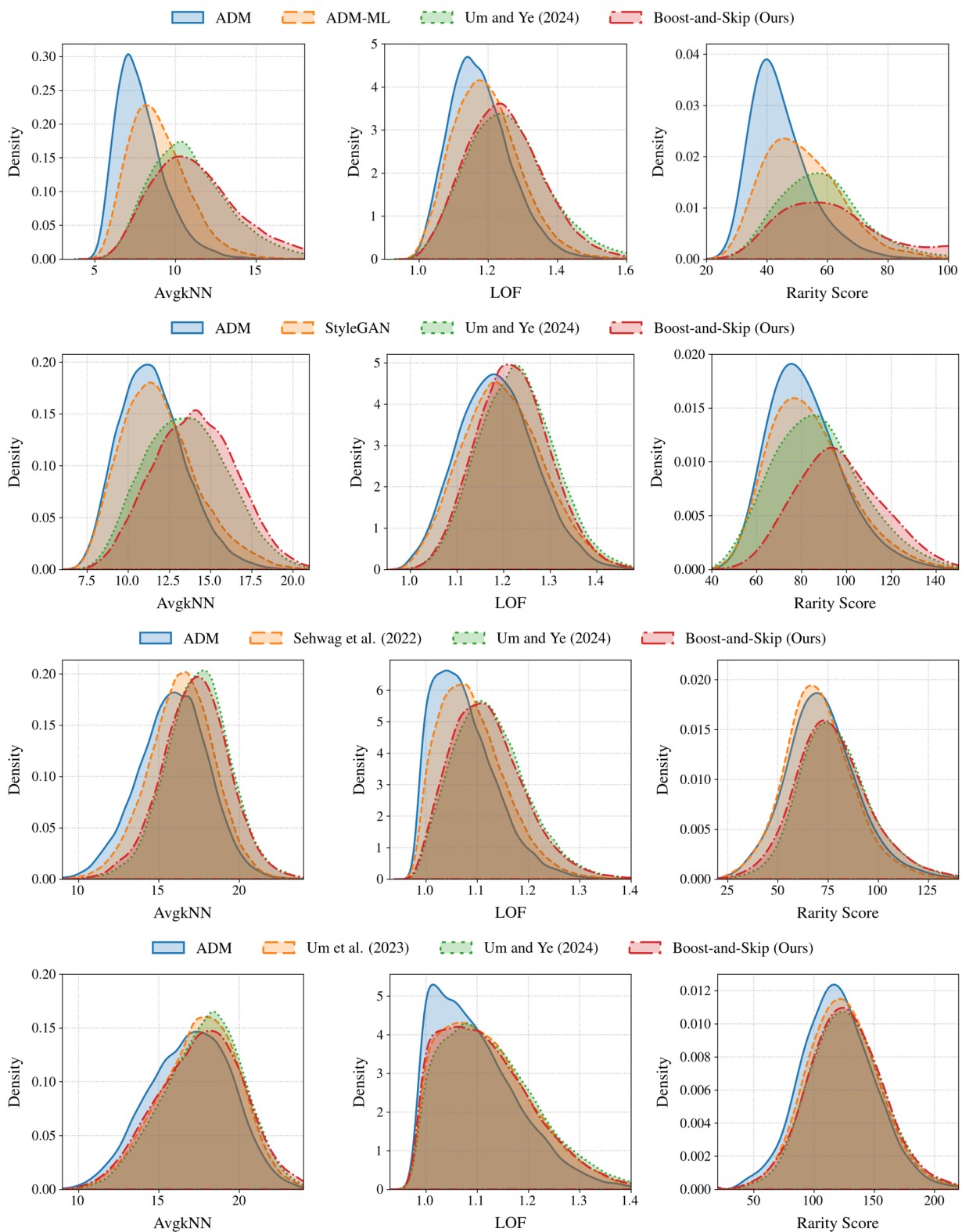

Figure 7: **Comparison of neighborhood density distributions across four benchmarks. (Top row)** CelebA $64 \times 64$. **(Second row)** LSUN-Bedrooms $256 \times 256$. **(Third row)** ImageNet $64 \times 64$. **(Fourth row)** ImageNet $256 \times 256$. "AvgkNN" refers to Average k-Nearest Neighbor, and "LOF" is Local Outlier Factor (Breunig et al., 2000). "Rarity Score" indicates a low-density metric proposed by Han et al. (2022). The higher values, the less likely samples for all three measures.

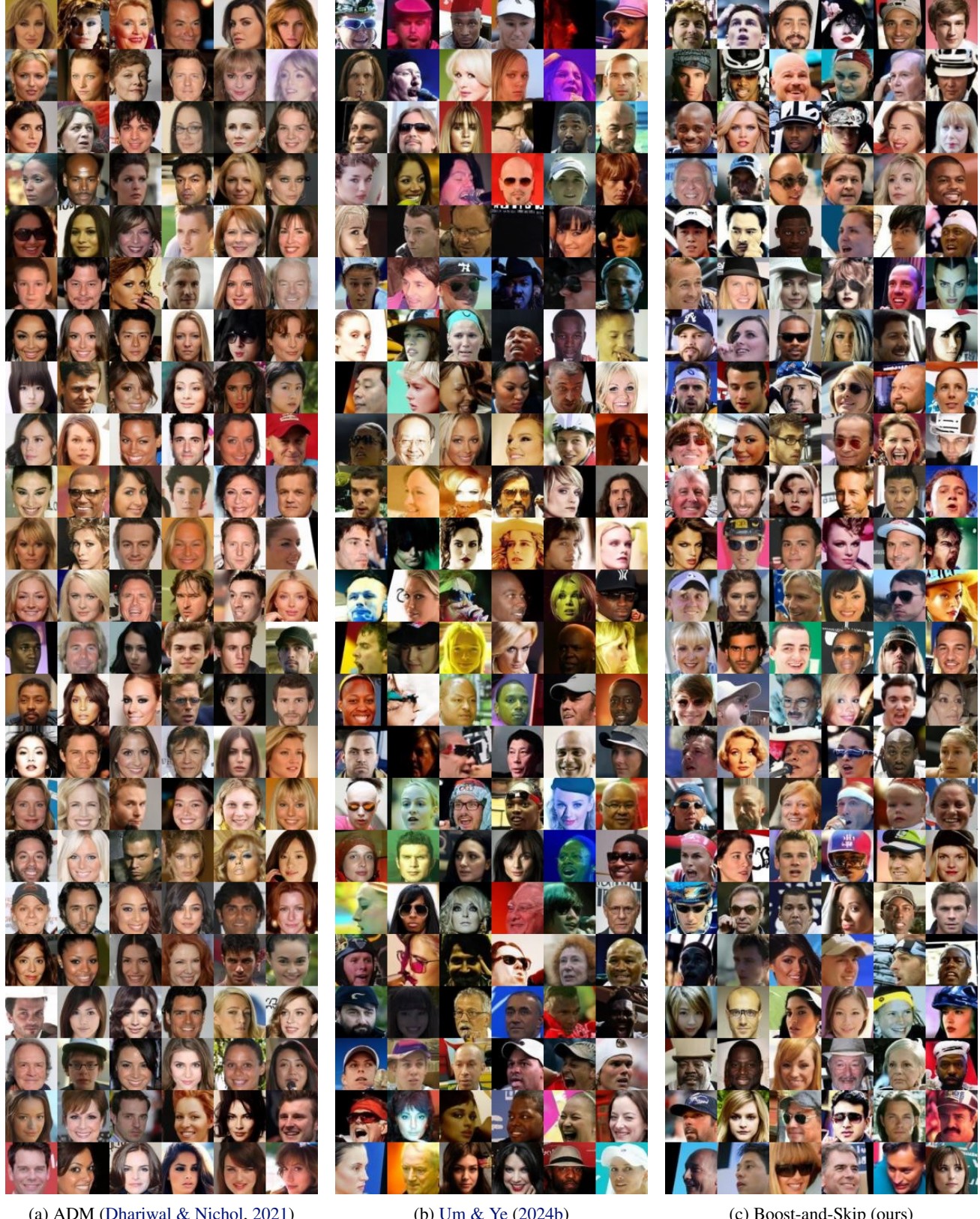

(a) ADM (Dhariwal & Nichol, 2021)    (b) Um & Ye (2024b)    (c) Boost-and-Skip (ours)

Figure 8: Samples comparison on CelebA $64 \times 64$.

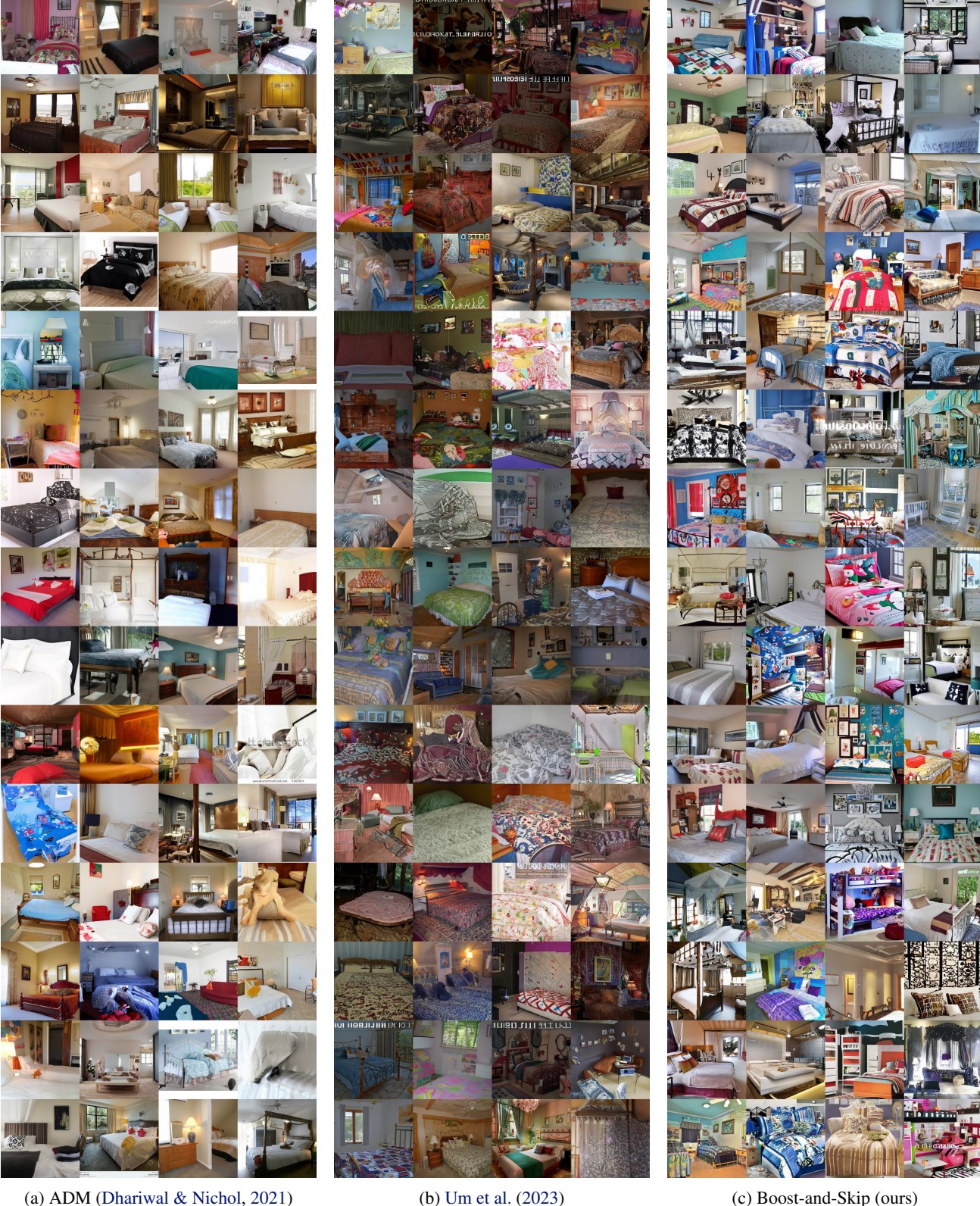

(a) ADM (Dhariwal & Nichol, 2021)  (b) Um et al. (2023)  (c) Boost-and-Skip (ours)

Figure 9: Samples comparison on LSUN-Bedrooms $256 \times 256$.

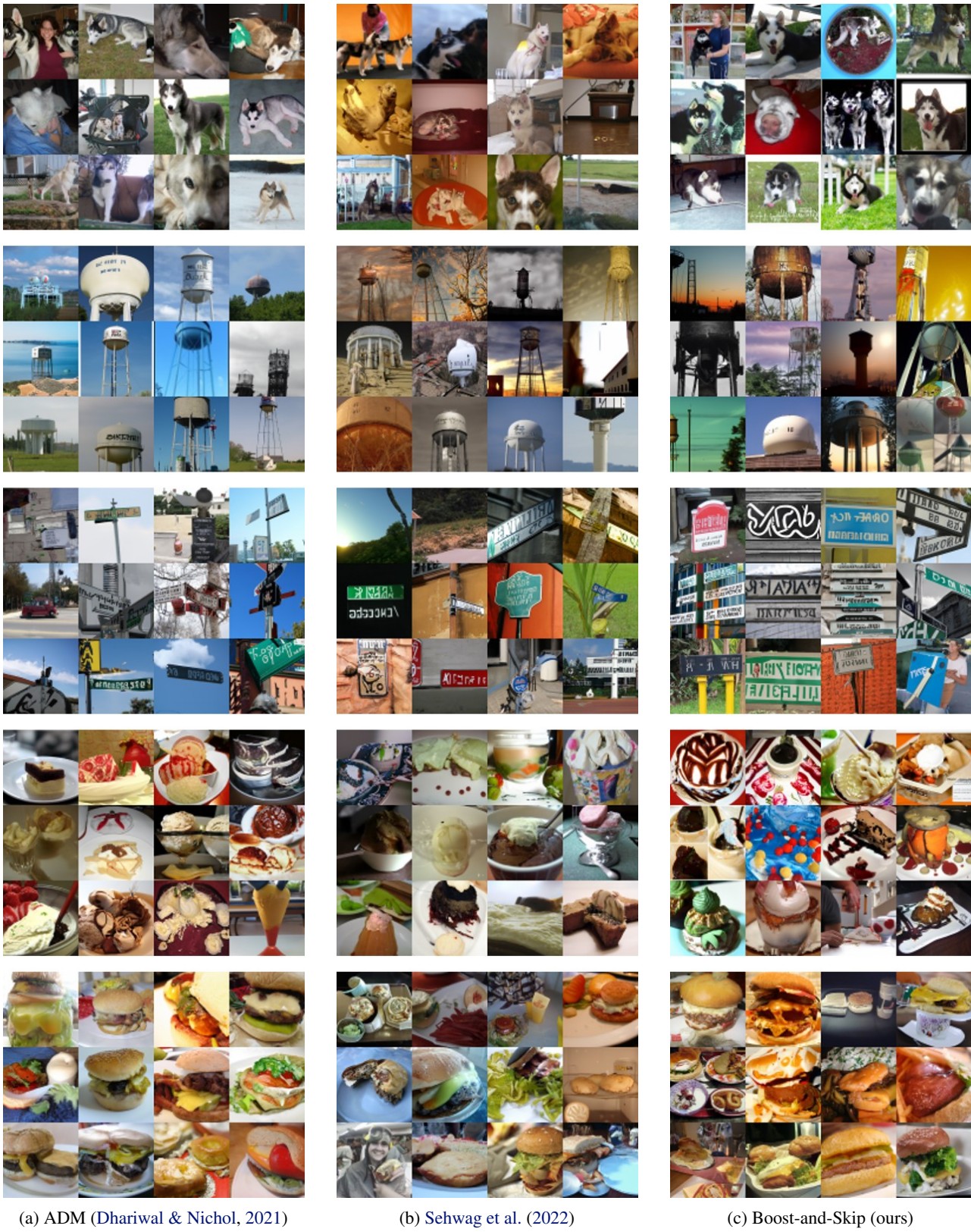

(a) ADM (Dhariwal & Nichol, 2021)          (b) Sehwag et al. (2022)          (c) Boost-and-Skip (ours)

Figure 10: Samples comparison on ImageNet $64 \times 64$. Generated samples from five classes are exhibited: (i) "Siberian husky" (top row); (ii) "water tower" (second row); (iii) "street sign" (third row); (iv) "ice cream" (fourth row); (v) "cheeseburger" (bottom row).

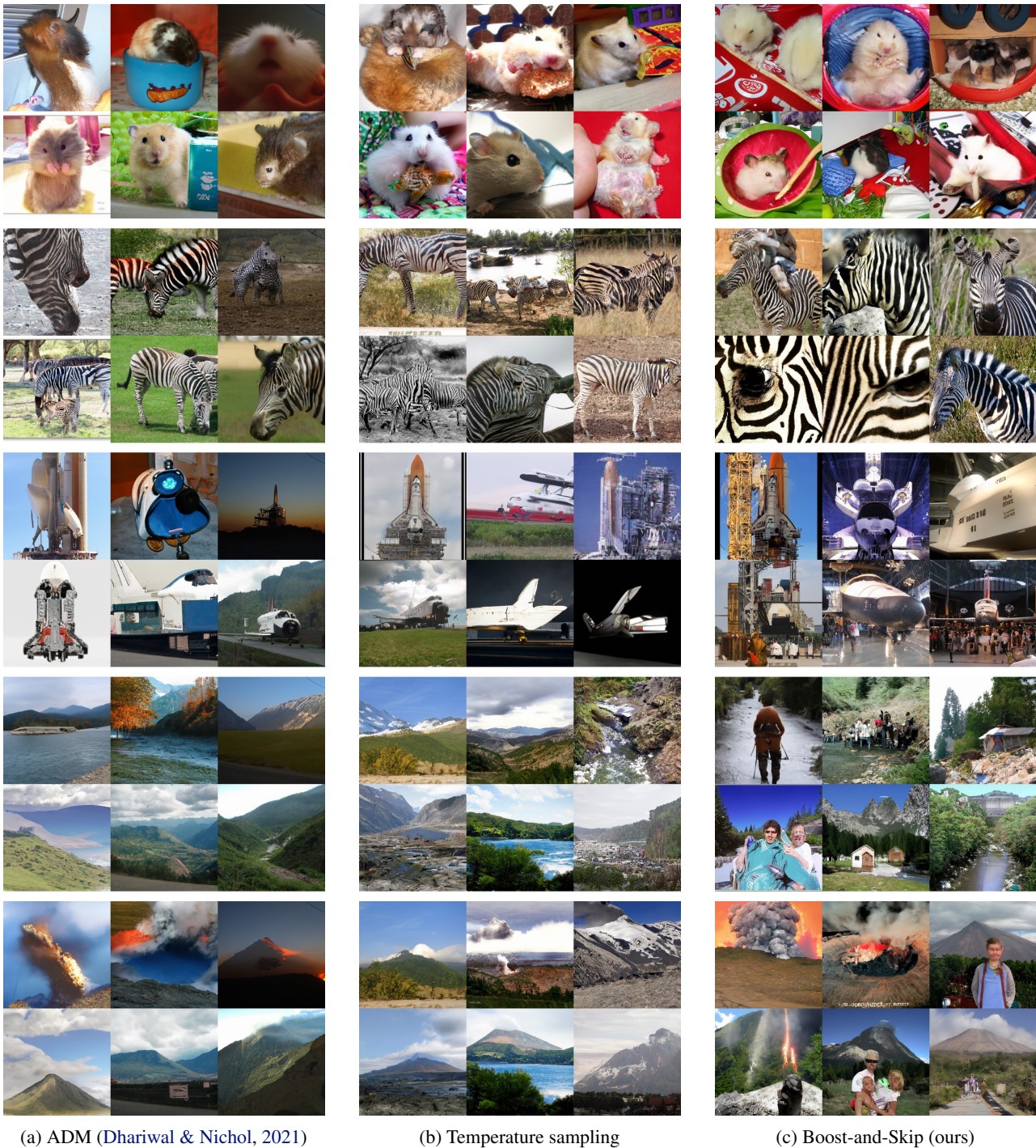

(a) ADM (Dhariwal & Nichol, 2021)  (b) Temperature sampling  (c) Boost-and-Skip (ours)

Figure 11: Samples comparison on ImageNet $256 \times 256$. Generated samples from five classes are exhibited: (i) "hamster" (top row); (ii) "zebra" (second row); (iii) "space shuttle" (third row); (iv) "valley" (fourth row); (v) "volcano" (bottom row).

