# OpenReview forum: "Boost-and-Skip: A Simple Guidance-Free Diffusion for Minority Generation"
_ICML.cc/2025/Conference — ICML 2025 poster_

### Official Review · Reviewer_o8Xg · 2025-03-13

**Overall Recommendation:** 5

**Summary:**

Authors of this work propose a method called "Boost-and-Skip" for generating minority samples from low-density regions in a data manifold using diffusion models. This method relies on two key modifications to the standard denoising process: 1) Initializing the reverse process with a higher variance noise (instead of a standard Gaussian noise), and 2) Skipping the early denoising steps. Unlike existing diffusion-based methods for minority generation, "Boost-and-Skip" does not rely on expensive guidance procedures increasing the efficiency. In addition, authors demonstrate that the proposed method achieves similar performance on the task of minority generation compared with state-of-the-art methods (diffusion-based and others) while maintaining the overall image quality, requiring no additional computations or modules, and being more efficient.

**Claims And Evidence:**

The main claims are that the proposed method is effective at generating minority samples with diffusion models while being more efficient than guidance-based methods. Both the theoretical and empirical results support these claims.

**Essential References Not Discussed:**

No. I found the related work section inclusive.

**Experimental Designs Or Analyses:**

I find the experimental design to be coherent with previous works on the task of minority generation. So, I do not see any particular pitfalls here.

**Methods And Evaluation Criteria:**

Authors followed previous works on the choice of metrics to report. I must admit that I am not an expert on this topic but I believe the reported metrics and evaluations do make sense for the problem of minority generation.

**Other Comments Or Suggestions:**

N/A

**Other Strengths And Weaknesses:**

Strengths:
1. Simple methodology that builds on two small modifications to the diffusion process for improving minority generation.
2. Efficient method that does not require additional components or training.
3. Achiving SOTA performance while reducing the inference time (as compared to SOTA methods).
4. Good write-up quality and ease of reading.

Weaknesses:
1. One of the main limitations of this work stems from its relies on the backbone diffusion model's exisiting biases. If the model has severe biases, then Boost-and-Skip will have limited affectiveness.
2. In addition, as discussed in Table 2, hyperparameters significantly influence the quality of generated samples. This suggests that effectiveness of the proposed method highly depends on a detailed exploration over the hyperparameter values, limiting the method's reliability.

**Questions For Authors:**

1. Can you discuss your point of view on the weaknesses I listed above and if and how these concerns can be addressed?
2. As can be seen the additional results included in Section D of appendix, it seems like the quality of generated samples is degraded when adjusting the base diffusion model for improved minority generation. For example, in Figure 8, it is easier to tell that samples in b and c are generated while samples in a are more realistic. Or in Figure 11, some samples are very weird-looking (humans in column c). I wonder if you know how badly can the sample quality be affected by improving the minority generation?

**Relation To Broader Scientific Literature:**

The minority generation problem is important on its own as it focuses on making generative models more inclusive and fair. Since the proposed method is a simple modification on top of standard diffusion models, it is a practical solution that has potential to be easily applied to various frameworks. I also really like the fact that the method builds on simple adjustments to existing processes. This can show that significant improvements are possible by minor smart changes and can encourage the research community to look for simple and efficient solutions for critical and challenging problems such as fairness.

**Theoretical Claims:**

I skimmed over all the equations included in the main paper and did not find any particular errors. However, I note that I did not do a careful read and I am not fully familiar with the background on some of the methodology details.

---

> ### Author Rebuttal · Authors · 2025-03-30
>
> We greatly appreciate Reviewer o8Xg for the strong acceptance and thoughtful feedback. Below, we provide detailed point-by-point responses to address your remaining concerns.
>
> ---
>
> > **1. [o8Xg] questioned the effectiveness of our approach on highly biased datasets.**
>
> To address your concern, we consider CIFAR10-LT, a highly-imbalanced version of CIFAR-10, and investigate the performance benefit of ours. See the table below for the results.
>
> | Method  | cFID  | sFID  | Prec | Rec |
> |---------|-------|-------|-----------|--------|
> | **DDPM** | 75.71 | 44.26 | 0.95      | 0.23   |
> | **CBGAN**| 78.62 | 43.76 | **0.99**      | 0.08   |
> | **BnS**  | **70.12** | **43.73** | 0.91      | **0.34**   |
>
> "CBGAN" refers to a class-balancing GAN approach that implements minority generation with minority conditional labels [1]. Similar to the experiments in our paper, we used real minority data from CIFAR10-LT as the reference for calculating the reported metrics. Observe that B&S outperforms the considered baselines (including the GAN-based approach in [1]) under this highly-biased benchmark, further demonstrating the robustness of our framework.
>
> ---
>
> > **2. [o8Xg] expressed concerns regarding the sensitivity to hyperparameters.**
>
> Although we acknowledge that our framework may be sensitive to hyperparameters — particularly $\gamma$ — we provide a practical heuristic for their selection in Appendix C (Lines 1250–1251), which streamlines the process of choosing both $\gamma$ and $\Delta_t$. Leveraging this approach, a simple one-dimensional grid search over $\gamma$ is sufficient to identify effective hyperparameters (see Lines 1267–1270 for details).
>
> We also highlight that the implementation complexity of our method is significantly lower than that of existing guided minority samplers, which often involve numerous design choices. For example, the approach in [2] requires training two separate classifiers with many design options (e.g., classifier architectures), while the method in [3] necessitates the selection of six hyperparameters. In contrast, our framework only requires choosing two hyperparameters, providing substantial practical benefits over the guided minority samplers in [2,3].
>
> ---
>
> > **3. [o8Xg] expressed concerns regarding the visual quality of minority samples.**
>
> We believe there are largely two reasons behind the quality degradation mentioned by the reviewer. First, there is a general trade-off between minority sampling performance and image quality. Second, the base diffusion model itself lacks the ability to generate minority features. We provide further explanation for each hypothesis below.
>
> To investigate the first hypothesis, since FID and Precision are measured with respect to ground-truth minority samples, we can use FID as a proxy for how close the generated distribution is to the distribution of minority data, and Precision as a proxy for the quality of generated minority samples. In Fig. 2 (provided in the link below), we observe there is generally a trade-off between the two quantities, and B&S provides competitive trade-off performance while achieving a dramatic reduction in inference cost compared to guidance-based minority methods. The reviewer is also directed to Table 2 (c), where we again observe FID vs. Precision trade-off as we adjust boosting strength in B&S.
>
> - Link to Fig. 2: https://docs.google.com/presentation/d/1dsMx8s5kJikQnjv6IQNvk-UyC_DgfU9xJpF-ZkaLweI/edit?usp=sharing
>
> Also, we hypothesize that in some cases, the base diffusion model may lack the capability to synthesize minority features. For instance, on ImageNet, it is well-known that generating human faces is challenging; see Fig. 7 in [4], Fig. 15 in [5], and Fig. 5 (b) in [2]. We believe that is why human faces in Fig. 11 (c) synthesized by B&S appear unnatural and distorted. Using a better base diffusion model may yield more realistic minority samples.
>
> ---
>
> **References**
>
> [1] Class Balancing GAN with a Classifier in the Loop, UAI 2021
>
> [2] Generating High Fidelity Data from Low-density Regions using Diffusion Models, CVPR 2022
>
> [3] Self-Guided Generation of Minority Samples Using Diffusion Models, ECCV 2024
>
> [4] Large Scale GAN Training for High Fidelity Natural Image Synthesis, ICLR 2019
>
> [5] Diffusion Models Beat GANs on Image Synthesis, NeurIPS 2021

---

> > ### Comment · Reviewer_o8Xg · 2025-04-04
> >
> > Thanks for the clarifications. I find the rebuttal effort by the authors helpful to answer my concerns and questions regarding the hyperparameter sensitivity and sample quality. I am happy to keep my original score.

---

> > > ### Author Response · Authors · 2025-04-05
> > >
> > > Thank you for your continued strong support and for acknowledging that our rebuttal addressed your concerns. We greatly appreciate your thoughtful evaluation and the time you dedicated to reviewing our work.

---

### Official Review · Reviewer_6VDS · 2025-03-14

**Overall Recommendation:** 3

**Summary:**

The paper proposes Boost-and-Skip, a method to generate low-density, minority samples. The method has two straightforward yet effective modifications to standard diffusion models: (1) variance-boosted initialization, and (ii) timestep skipping during the generative process. The authors provide intuitions, theoretical analysis, and synthetic experiments to motivate these two modifications. Empirically, they show that their method achieves competitive performance compared to state-of-the-art guidance-based methods but at significantly lower computational costs.

**Claims And Evidence:**

The claims made by the authors (i.e., variance-boosted initialization and timestep skipping) are clearly stated and supported by experiments.

**Essential References Not Discussed:**

N/A

**Experimental Designs Or Analyses:**

The experimental design is rigorous. The ablation studies demonstrate the necessity and impact of each component of the proposed approach (variance-boosting and timestep skipping).

**Methods And Evaluation Criteria:**

The proposed variance-boosted initialization and timestep-skipping are straightforward and intuitive. The evaluation criteria (e.g., cFID) and datasets (e.g., CelebA, ImageNet) are standard and appropriate for assessing minority generation performance.

**Other Comments Or Suggestions:**

N/A

**Other Strengths And Weaknesses:**

N/A

**Questions For Authors:**

N/A

**Relation To Broader Scientific Literature:**

The paper is well related to previous works on diffusion models and minority generation. It clearly addresses the efficiency limitation of existing guidance-based approaches and provides an effective alternative.

**Theoretical Claims:**

The paper includes theoretical claims concerning the properties of their method. I checked the claims and found that they help me better understand the intuition behind the method; however, I didn't check the correctness of the claims.

---

> ### Author Rebuttal · Authors · 2025-03-30
>
> Thank you for your thoughtful feedback and for considering our work for acceptance. We appreciate your time and evaluation. If you have any further suggestions or questions, feel free to let us know. We are more than happy to address any additional points or provide further clarifications as needed.

---

### Official Review · Reviewer_7UiG · 2025-03-14

**Overall Recommendation:** 4

**Summary:**

The paper proposes an approach called Boost-and-Skip for generating minority samples using diffusion models. Specifically, it begins stochastic generation with variance-boosted noise to encourage initializations in low-density regions. It then skips several of the earliest timesteps to further amplify the impact of low-density initialization. The effectiveness of Boost-and-Skip is supported by both theoretical and empirical evidence, with the added advantage of low computational cost.

**Claims And Evidence:**

The claims are clear and well supported by theoretical and empirical evidence.

**Essential References Not Discussed:**

The literature on conditional diffusion models should be reviewed in the main text of the paper, as this is another important methodological branch for minority generation.

**Experimental Designs Or Analyses:**

The empirical study is extensive, covering multiple benchmark datasets and baselines.

**Methods And Evaluation Criteria:**

The proposed Boost-and-Skip is quite simple, yet its advantage lies in its rigorous theoretical support.

Some issues:

1. I feel that the proposed method is sensitive to hyperparameters, requiring a wide range of grid searches. Moreover, the optimal hyperparameter settings vary significantly across different datasets. Table 2(c) presents the substantial differences in results under different hyperparameter settings.

2. The authors consider baselines beyond diffusion models. I think some GAN works specifically focused on minority generation should be considered, rather than general-purpose GAN works.

**Other Comments Or Suggestions:**

N.A.

**Other Strengths And Weaknesses:**

The writing in this paper is clear and easy to follow.

**Questions For Authors:**

1. From Fig. 7, it appears the proposed method would generate OOD samples (original density0). This could lead to an inaccurate learned distribution. How do you explain this limitation?

2. I think the learned distribution for the toy data (Fig. 2) can be plotted to evaluate how well it corresponds to the theoretical result in Proposition 3.2.

3. I wonder about the effectiveness of the proposed method on extremely imbalanced problems. The setup in Figure 5 does not clearly demonstrate this. I suggest, for example, using the CelebA dataset with 99% young and 1% old faces. Additionally, a comparison with conditional diffusion models, given known conditional labels, should be included.

4. The sensitivity to hyperparameters (Methods And Evaluation Criteria).

**Relation To Broader Scientific Literature:**

The paper proposes a simple approach for generating minority samples using diffusion models, achieving performance comparable to state-of-the-art baselines with lower complexity.

**Theoretical Claims:**

I did not check the proof very carefully. But their theoretical claims look reasonable.

---

> ### Author Rebuttal · Authors · 2025-03-30
>
> We appreciate the reviewer for your detailed comments and valuable suggestions. Below, we provide thorough point-by-point responses to address your concerns.
>
> ---
>
> > **1. [7UiG] expressed a concern on the sensitivity to hyperparameters.**
>
> Please refer to our response to Reviewer o8Xg (the second bullet point).
>
> ---
>
> > **2. [7UiG] suggested comparisons with GAN-based minority generation frameworks.**
>
> To reflect your comment, we have conducted new experiments to compare our method with GAN-based minority generation frameworks. Specifically, we evaluate a class-balancing GAN approach [1] and compare its performance in generating minority samples with ours on CIFAR10-LT (a long-tailed version of CIFAR-10). See the table below for the results.
>
> | Method  | cFID  | sFID  | Prec | Rec |
> |---------|-------|-------|-----------|--------|
> | **DDPM** | 75.71 | 44.26 | 0.95      | 0.23   |
> | **CBGAN**| 78.62 | 43.76 | **0.99**      | 0.08   |
> | **BnS**  | **70.12** | **43.73** | 0.91      | **0.34**   |
>
> As in the experiments presented in our paper, we used real minority data (from CIFAR10-LT) as the reference for computing the metrics. Observe that B&S outperforms the GAN-based approach in [1] (i.e., CBGAN) even under this highly-biased benchmark, further highlighting its effectiveness as a minority generator.
>
> ---
>
> > **3. [7UiG] pointed out that literature reviews on minority-conditional diffusion models are missing.**
>
> We kindly remind the reviewer that approaches with minority-conditional diffusion models are discussed in the related work section in Appendix A.1. We will move them to the main body in our revision.
>
> ---
>
> > **4. Clarifications on Fig. 7.**
>
> We kindly remind the reviewer that the focus of minority generation is not to replicate the training data distribution but to intentionally bias generation toward minority instances, which are defined as low-density on-manifold samples. In this regard, Fig. 7 demonstrates that our framework effectively achieves this goal, as the high-valued neighborhood metric values imply the generation of low-density instances.
>
> While the reviewer might be concerned that high neighborhood metric values indicate the presence of off-manifold (i.e., OOD) samples of poor quality, we emphasize that our method does not generate more OOD samples than state-of-the-art minority samplers such as Minority Guidance [2]. This is evidenced by our superior FID scores compared to Minority Guidance (e.g., in Table 1), where FID is computed using real minority (i.e., on-manifold, low-density) data.
>
> ---
>
> > **5. [7UiG] suggested a sanity check of Proposition 3.2 using the toy data in Fig. 2.**
>
> Per your suggestion, we did an experiment for the sanity check in Fig. 1 (provided in the link below). Specifically, we plot generated data variance as a function of initial Gaussian noise variance for the two rings example. Blue dotted line denotes generated data variance predicted by theory, and the orange solid line illustrates actual generated data variance. The general trend of the two curves are similar, validating the ability of B&S to generate minority samples. We conjecture that the offset between theory and practice may arise from the score estimation error, as we use learned scores rather than exact scores to simulate B&S.
>
> - Link to Fig. 1: https://docs.google.com/presentation/d/1GCZKcxbX_A7e_v_ckVCefcVxGsbm2UkZSxyUc5zuSbA/edit?usp=sharing
>
> ---
>
> > **6. [7UiG] questioned the effectiveness of our approach on highly imbalanced benchmarks.**
>
> To address your question, we consider CIFAR10-LT, a highly-imbalanced version of CIFAR-10, and explore the performance benefit of ours. We found that B&S yields improved minority generation even under this biased setting; see the table above (included in the second bullet point of this response) for detailed results.
>
> ---
>
> > **7. [7UiG] suggested comparisons with minority-conditional diffusion frameworks.**
>
> We gently remind the reviewer that our experiments already encompass such approaches, by incorporating ADM-ML [2] - a classifier-guided diffusion sampler conditioned on known minority labels in CelebA. See details in Table 1.
>
> ---
>
> **References**
>
> [1] Class Balancing GAN with a Classifier in the Loop, UAI 2021
>
> [2] Don’t Play Favorites: Minority Guidance for Diffusion Models, ICLR 2024

---

> > ### Comment · Reviewer_7UiG · 2025-04-03
> >
> > Thanks for the detailed responses. My concerns have been addressed.
> >
> > Just a small point—the comparison of conditional diffusion I mentioned refers to the class-free version. Anyway, I vote for the acceptance of this work after the rebuttal.

---

> > > ### Author Response · Authors · 2025-04-03
> > >
> > > Thank you for raising the score. We are pleased that our rebuttal successfully addressed your previous concerns. Per your suggestion, we will include more comparisons with conditional diffusion frameworks, including some classifier-free versions.

---

### Official Review · Reviewer_bs5F · 2025-03-17

**Overall Recommendation:** 3

**Summary:**

The paper provides two techniques for improving minority sampling. The first technique is a boost, which initialises the sampling with controllable variance. The second technique is skip where it will skip several sampling timesteps. The authors claim to achieve better performance in generating minority samples.

**Claims And Evidence:**

The evidence for achieving a minority sample by the proposed method is not very clear both qualitatively and quantitatively

**Essential References Not Discussed:**

The work does not compare with guidance methods to show why this method is better than guidance.

**Experimental Designs Or Analyses:**

experimental designs are okey

**Methods And Evaluation Criteria:**

1. The method is straightforward and intuitive

2. The improvement is mainly compared with Temperature sampling. Most of the time, the performance is poorer on the baselines (Table 1). Please add more explanation

3. It is hard to judge Figure 5. Can not tell why boost-and-skip provides better minority samples

4. Lack of clear metrics to measure minority samples. In the paper, the authors mention AvgkNN, LOF and Rarity Score, yet all the tables do not have these figures.

**Other Comments Or Suggestions:**

n/a

**Other Strengths And Weaknesses:**

n/a

**Questions For Authors:**

1. The improvement is mainly compared with Temperature sampling. Most of the time, the performance is poorer on the baselines (Table 1). Please add more explanation

2. The work does not compare with guidance methods to show why this method is better than guidance. Please include more comparisons.

3. Add more metrics about the minority sample. Current metrics in the paper does not show how minority samples are covered.

**Relation To Broader Scientific Literature:**

n/a

**Theoretical Claims:**

I checked the theoretical claims but not sure if the proofs are correct.

---

> ### Author Rebuttal · Authors · 2025-03-30
>
> We thank Reviewer bs5F for the constructive feedback. Below we provide point-by-point responses on your questions and concerns.
>
> ---
> > **1. [bs5F] expressed a concern that the performance is often limited compared to baselines.**
>
> We note that the superior baselines in Table 1 (e.g., [1,2]) correspond to guidance-based minority approaches that employ guidance terms to direct inference toward low-density regions. While we acknowledge that our suboptimal performance may stem from the lack of such explicit guidance for minority generation, our method significantly advances the Pareto frontier in the performance-complexity tradeoff (see Fig. 1). In particular, our framework delivers notable computational benefits over the guided minority approaches. For example, on ImageNet-64, our method achieves **65% reduction in wall-clock time and 4.5× lower peak memory usage** compared to the current state-of-the-art [2] (see Table 3).
>
> ---
>
> > **2. Clarifications on Fig. 5.**
>
> We would like to assure the reviewer that the visual attributes of our samples in Fig. 5 capture distinctive features of minority instances. For instance, jack-o’-lantern images with bright surroundings (rather than the typical dark Halloween atmosphere) are considered as low-density features of the class [3]. Also, our eagle-class images in the same figure exhibit more intricate visual details compared to the baselines, which are also known as minority features [1,2].
>
> ---
>
> > **3. [bs5F] noted missing neighborhood metrics like AvgkNN.**
>
> As noted in L356-358 (right column), the evaluation results using AvgkNN, LOF, and Rarity Score are provided in Appendix D.1, where B&S performs consistently well across all three metrics, rivaling the state-of-the-art guided minority sampler [2]. See Fig. 7 therein for explicit details.
>
> ---
>
> > **4. [bs5F] noted missing comparisons with guidance-based minority methods.**
>
>
> We would like to gently remind the reviewer that, in Tables 1,3,4, we already compare B&S with guidance-based methods such as ADM-ML [1], Minority Guidance [1], and Self-guidance for minority generation [2].
>
> ---
>
> **References**
>
> [1] Don’t Play Favorites: Minority Guidance for Diffusion Models, ICLR 2024
>
> [2] Self-Guided Generation of Minority Samples Using Diffusion Models, ECCV 2024
>
> [3] Generating High-Fidelity Data from Low-Density Regions Using Diffusion Models, CVPR 2022

---

### Decision · Program_Chairs · 2025-05-01

**Decision:**

Accept (poster)

**Comment:**

This paper proposes a new method, Boost-and-Skip (B&S), for minority sample generation using diffusion models. With only minimal modifications—variance boosting and timestep skipping—the method significantly improves minority generation performance without relying on guidance, while also reducing computational cost.

Reviewers appreciated the theoretical motivation and empirical effectiveness of the approach. The authors provided clear and sufficient responses to the concerns raised. The simplicity and general applicability of the method are also notable strengths.

Overall, this paper presents an effective and efficient approach to minority sample generation and is recommended for acceptance.